# MEMBERSHIP INFERENCE ATTACKS AGAINST FINE-TUNED DIFFUSION LANGUAGE MODELS

**Yuetian Chen**[1], **Kaiyuan Zhang**[1], **Yuntao Du**[1], **Edoardo Stoppa**[1],
**Charles Fleming**[2], **Ashish Kundu**[2], **Bruno Ribeiro**[1], **& Ninghui Li**[1]

[1]Department of Computer Science, Purdue University

[2]Cisco Research

`{yuetian, zhan4057, ytdu, estoppa, ribeirob, ninghui}@purdue.edu`
`{chflemin, ashkundu}@cisco.com`

## ABSTRACT

Diffusion Language Models (DLMs) represent a promising alternative to autoregressive language models, using bidirectional masked token prediction. Yet their susceptibility to privacy leakage via Membership Inference Attacks (MIA) remains critically underexplored. This paper presents the first systematic investigation of MIA vulnerabilities in DLMs. Unlike the autoregressive models' single fixed prediction pattern, DLMs' multiple maskable configurations exponentially increase attack opportunities. This ability to probe many independent masks dramatically improves detection chances. To exploit this, we introduce SAMA (**S**ubset-**A**ggregated **M**embership **A**ttack), which addresses the sparse signal challenge through robust aggregation. SAMA samples masked subsets across progressive densities and applies sign-based statistics that remain effective despite heavy-tailed noise. Through inverse-weighted aggregation prioritizing sparse masks' cleaner signals, SAMA transforms sparse memorization detection into a robust voting mechanism. Experiments on nine datasets show SAMA achieves 30% relative AUC improvement over the best baseline, with up to $8\times$ improvement at low false positive rates. These findings reveal significant, previously unknown vulnerabilities in DLMs, necessitating the development of tailored privacy defenses.

## 1 Introduction

Large Language Models (LLMs) demonstrate transformative capabilities across diverse applications (Team, 2024; Qwen, 2025; DeepSeek-AI, 2025). While autoregressive models (ARMs) based on sequential next-token prediction remain dominant, Diffusion Language Models (DLMs) are rapidly emerging as a compelling alternative (Gulrajani & Hashimoto, 2023; Gong et al., 2024; Ye et al., 2025b). These models, leveraging bidirectional context to reverse data corruption by reconstructing masked tokens, are increasingly adopted for their strong scalability, performance competitive with ARM counterparts, and their ability to address ARM-specific limitations like the reversal curse (Berglund et al., 2024). Recent industrial developments, such as Google's Gemini Diffusion (DeepMind, 2025), further underscore the growing relevance of this paradigm, offering performance competitive with significantly larger autoregressive models.

The privacy risks associated with LLMs, particularly their propensity to memorize training data, have been extensively studied for ARMs (Carlini et al., 2021; Mireshghallah et al., 2024; Merity et al., 2018; Hartmann et al., 2023). Membership Inference Attacks (MIAs) (Shokri et al., 2017), which aim to determine if a specific data sample was part of a model's training corpus, are the primary benchmark for quantifying these risks, with numerous techniques developed to probe ARM vulnerabilities (Mattern et al., 2023; Duan et al., 2024; Fu et al., 2024). However, for the burgeoning class of DLMs, which operate on distinct principles of *iterative, bidirectionally-conditioned mask prediction*, the extent and nature of such privacy vulnerabilities remain critically uncharted. Their unique architectural properties deviate significantly from ARMs, rendering existing MIA strategies

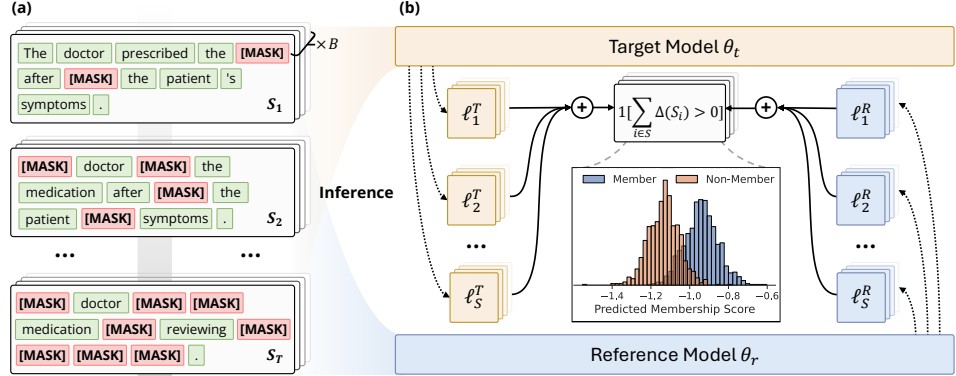

Figure 1: An overview of the SAMA. **(a)** An input sequence `The` `doctor` `prescribed` `the` `medication` `after` `reviewing` `the` `patient` `'s` `symptoms` `.` undergoes progressive masking over $S$ steps, accumulating masked positions. **(b)** The target and reference DLMs' reconstruction errors for masked tokens `[MASK]` are tracked position-wise at each step ($\{\ell^s\}_{s=1}^S$). Sign-based test statistics detect when specific mask configurations activate memorization, computing the fraction of configurations where reference loss exceeds target loss. This sign-based comparison, combined with inverse weighting, forms the membership score to differentiate training members from non-members.

potentially ill-suited. The privacy implications of these distinctions have been largely overlooked. To the best of our knowledge, this work presents the first systematic investigation into the privacy risks of emerging DLMs through the lens of MIAs.

The absence of privacy analysis focused on DLMs constitutes an important gap and raises a key question: *Do the unique paradigms of DLMs introduce novel vulnerabilities exploitable by MIAs?* We focus on fine-tuning, where such privacy risks are typically amplified (Mireshghallah et al., 2022a; Fu et al., 2024; Huang et al., 2025; Meng et al., 2025). In this work, we address this question through three key contributions:

1. We analyze how DLMs' multiple masking configurations, unlike the single fixed pattern in ARMs, change the membership inference problem. The ability to probe many independent masks exponentially increases the chances of detecting memorization patterns learned during fine-tuning.

2. We empirically validate this heightened susceptibility and effectively exploit these unique characteristics by designing SAMA (**S**ubset-**A**ggregated **M**embership **A**ttack), an MIA framework that combines progressive masking with robust sign-based statistics—addressing the fundamental challenge that only sparse mask configurations reveal membership while most yield noise.

3. Through comprehensive experiments on state-of-the-art DLMs (i.e., LLaDA-8B-Base (Nie et al., 2025), and Dream-v0-Base-7B (Ye et al., 2025a)) across diverse datasets (i.e., six different domain splits in MIMIR (Duan et al., 2024), WikiText-103 (Merity et al., 2016), AG News (Zhang et al., 2015), and XSum (Narayan et al., 2018)), we demonstrate that SAMA significantly outperforms a wide range of existing MIA baselines, thereby uncovering and quantifying critical privacy vulnerabilities.

The high-level overview of how SAMA works is shown in Figure 1, by systematically sampling diverse mask configurations across progressively increasing densities, then applying sign tests robust to sparse signals, we transform the detection problem from finding large signals in noise to identifying consistent patterns across multiple independent probes.

## 2 Preliminaries

This section describes the threat model under which SAMA operates, and provides background on DLMs, focusing on the mechanisms relevant to our work.

## 2.1 Threat Model

This paper investigates MIAs (Shokri et al., 2017) targeting DLMs, focusing on vulnerabilities introduced during fine-tuning. Throughout this work, we use subscripts DF and AR to denote diffusion and autoregressive models, respectively. The adversary aims to determine whether a specific text sequence $\mathbf{x} = \{x_i\}_{i=1}^{L} \in \mathcal{V}^L$, where $\mathcal{V}$ is the model's token vocabulary and $L$ is the sequence length, was part of the training dataset $D_{\text{train}}$ of a target model $\mathcal{M}_{\text{DF}}^{\text{T}}$. This is formalized as inferring the binary membership status $\mu(\mathbf{x}) \in \{0, 1\}$, where $\mu(\mathbf{x}) = 1$ indicates $\mathbf{x} \in D_{\text{train}}$. The adversary constructs an attack function $A(\mathbf{x}, \mathcal{M}_{\text{DF}}^{\text{T}})$ that outputs a membership prediction $\hat{\mu}(\mathbf{x})$.

We assume a *grey-box access model*, where the adversary can query the target model $\mathcal{M}_{\text{DF}}^{\text{T}}$ with arbitrary inputs, including custom partially masked sequences. In response, the adversary receives detailed outputs such as predictive probability distributions or logits for specified token positions. However, the adversary cannot access internal parameters, gradients, or activations. Following reference-based MIA approaches (Watson et al., 2021; Mireshghallah et al., 2022a; Fu et al., 2024; Huang et al., 2025; Meng et al., 2025), the adversary also has grey-box access to a reference model $\mathcal{M}_{\text{DF}}^{\text{R}}$. The ideal reference is the pre-trained base model from which $\mathcal{M}_{\text{DF}}^{\text{T}}$ was derived, as this best isolates fine-tuning-specific memorization. When unavailable, alternative reference models can be used, with implications discussed in Section D.6. Practically, this access model aligns with current open-weight deployment practices where fine-tuned weights are publicly released alongside base models. Furthermore, emerging DLM serving interfaces, such as LLaDA's in-filling demos (Hugging Face), inherently support the custom masked inputs required for this threat model.

## 2.2 Diffusion Language Models

While autoregressive models predicting tokens sequentially represent the dominant LLM paradigm (Radford et al., 2019; Brown et al., 2020), DLMs offer a distinct approach, gaining prominence in the language domain (Austin et al., 2023; Lou & Ermon, 2023; Shi et al., 2024a; Ou et al., 2024). These models typically learn to reverse a data corruption process, commonly implemented via iterative masking and prediction. This process starts with an original sequence $\mathbf{x}$ of length $L$, where randomly selected tokens are replaced by a special [MASK] token. We denote the set of masked positions as $\mathcal{S} \subseteq [L]$, where $[L] = \{1, 2, \cdots, L\}$. The model is then trained to predict the original tokens at positions in $\mathcal{S}$ given the partially masked sequence.

**Training Procedure.** A notable example of DLM is LLaDA (**L**arge **La**nguage **D**iffusion with m**A**sking) (Nie et al., 2025); we use its paradigm to detail the training formulation. The central component is a mask predictor, usually a Transformer architecture, which learns to predict the original tokens $x_i$ at the masked positions (i.e., $i \in \mathcal{S}$), conditioned on the unmasked context $\mathbf{x}_{-\mathcal{S}} := \{x_i : i \in [L] \setminus \mathcal{S}\}$. The model is optimized by minimizing the cross-entropy loss computed on mini-batches from the training dataset $D_{\text{train}}$. For each training sequence $\mathbf{x} \in D_{\text{train}}$ and randomly sampled mask configuration $\mathcal{S}$, the loss is:

$$\ell(\mathbf{x}, \mathcal{S}) = -\frac{1}{|\mathcal{S}|} \sum_{i \in \mathcal{S}} \log p_{\mathcal{M}}(x_i | \mathbf{x}_{-\mathcal{S}}), \tag{1}$$

where $p(x_i | \mathbf{x}_{-\mathcal{S}})$ denotes the probability assigned by model $\mathcal{M}$ to token $x_i$ at position $i$, conditioned on the unmasked context $\mathbf{x}_{-\mathcal{S}}$, and $\mathcal{S}$ is sampled with varying cardinalities to ensure the model learns to reconstruct tokens under different masking densities. This training objective relies on random sampling across the entire configuration space, where the model sees diverse, unpredictable masking patterns during training. This objective serves as a variational bound on the negative log-likelihood, grounding it in established generative modeling principles.

## 3 SAMA: Exploiting Bidirectional Dependencies in DLMs

DLMs process text through bidirectional masking mechanisms rather than the unidirectional approach of autoregressive models. This architectural distinction creates fundamentally different memorization patterns during fine-tuning, which we investigate for membership inference vulnerabilities through our proposed SAMA.

## 3.1 Membership Signals in Autoregressive vs. Diffusion Models

We formalize how the architectural differences between ARMs and DLMs affect membership signals, focusing on how each model type constrains the attacker's ability to probe for memorization. For a text sequence $\mathbf{x}$, we define the membership signals as the loss difference between a reference model $\mathcal{M}^R$ and a fine-tuned target model $\mathcal{M}^T$. This difference captures memorization patterns introduced during fine-tuning.

In ARMs, the membership signals are deterministic and fixed by the autoregressive factorization:

$$\Delta_{\text{AR}}(\mathbf{x}) = \ell_{\text{AR}}(\mathbf{x}; \mathcal{M}_{\text{AR}}^R) - \ell_{\text{AR}}(\mathbf{x}; \mathcal{M}_{\text{AR}}^T) = \frac{1}{L} \sum_{i=1}^{L} \left[ \ell_i^R(\mathbf{x}) - \ell_i^T(\mathbf{x}) \right], \qquad (2)$$

where $\ell_i = -\log p(x_i|x_{<i})$ represents the loss at position $i$ given only the preceding context. Crucially, this provides exactly one context configuration per token position—the attacker observes only left-to-right dependencies, making any backward or bidirectional relationships invisible to membership inference.

In DLMs, the membership signals depend on the chosen mask configuration $\mathcal{S} \subseteq [L]$:

$$\Delta_{\text{DF}}(\mathbf{x}; \mathcal{S}) = \ell_{\text{DF}}(\mathbf{x}; \mathcal{S}, \mathcal{M}_{\text{DF}}^R) - \ell_{\text{DF}}(\mathbf{x}; \mathcal{S}, \mathcal{M}_{\text{DF}}^T) = \frac{1}{|\mathcal{S}|} \sum_{i \in \mathcal{S}} \left[ \ell_i^R(\mathbf{x}, \mathcal{S}) - \ell_i^T(\mathbf{x}, \mathcal{S}) \right], \qquad (3)$$

where $\ell_i(\mathbf{x}, \mathcal{S}) = -\log p(x_i|\mathbf{x}_{-\mathcal{S}})$ represents the loss for predicting token $i$ given all unmasked tokens in sequence $\mathbf{x}$.

This DLM mask-configuration dependency creates both challenges and opportunities for membership inference. The challenge arises because an arbitrary mask configuration $\mathcal{S}$ chosen during inference may not align with the masking patterns the model saw during training, resulting in weak or noisy signals. During fine-tuning, the DLM memorizes specific token relationships under particular masking patterns; when a chosen configuration aligns even partially with these training patterns (e.g., sharing key masked positions or preserving critical context tokens), the memorization activates and the signals $\Delta_{\text{DF}}(\mathbf{x}; \mathcal{S})$ becomes large. Conversely, configurations rarely or never seen during training yield weak signals dominated by noise.

However, this same property creates two distinct advantages for membership inference in DLMs. First, we gain multiple independent probing opportunities. Rather than extracting a single fixed signal $\Delta_{\text{AR}}(\mathbf{x})$, we can collect a set of configuration-dependent signals $\{\Delta_{\text{DF}}(\mathbf{x}; \mathcal{S}_1), \Delta_{\text{DF}}(\mathbf{x}; \mathcal{S}_2), \ldots\}$, and if we sample enough diverse masks, we may uncover stronger membership signals than available in ARMs. Second, the bidirectional context enables probing token relationships impossible in ARMs. Consider tokens $x_i$ and $x_j$ with a strong semantic relationship memorized during fine-tuning: in an ARM, if $j > i$, this relationship can never influence the prediction of $x_i$ since $x_j$ is always masked, but in a DLM, we can probe this relationship through multiple configurations by masking only $\{x_i\}$ to see how $x_j$ influences its prediction, masking only $\{x_j\}$ for the reverse, or masking both $\{x_i, x_j\}$ to test their joint reconstruction. Each configuration potentially reveals different memorization patterns inaccessible to unidirectional models.

To validate the hypothesis that membership signals in DLMs are related to the mask configuration, we conducted a controlled experiment on the ArXiv dataset. For a balanced set of member and non-member samples, we evaluated 100 distinct random mask configurations per sample and analyzed the resulting distribution of signal strengths $\Delta_{DF}(x; \mathcal{S})$. Figure 2 presents these findings. Panel (a) illustrates the aggregate signal densities, showing that while non-member signals form a symmetric distribution centered near zero, member signals exhibit a distinct positive shift with a heavier right tail. Panel (b) reveals the high intra-sample variance through violin plots of individual samples. The variance in signal strength caused simply by changing the mask configuration ($\sigma \approx 0.10$) exceeds the average margin between members and non-members ($\delta \approx 0.06$). This demonstrates that the membership signal is not a static property of the input text but fluctuates significantly depending on whether the mask hides specific "memorable" tokens. Consequently, a single random mask is statistically likely to miss the sparse membership signal, necessitating a multi-configuration probing strategy to robustly distinguish members from non-members.

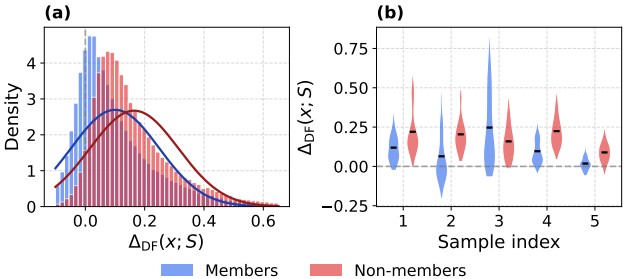

Figure 2: Empirical analysis of signal sparsity and configuration dependency. **(a)** The aggregate density of signal strengths $\Delta_{DF}(x; \mathcal{S})$ shows that member signals (red) are shifted positively compared to the more zero-centered non-member noise (blue), yet possess significant overlap. **(b)** Violin plots for individual samples reveal that the intra-sample variance, fluctuations in signal strength caused solely by changing the mask configuration, is substantial. This high variance confirms that membership signals are sparse and configuration-dependent, motivating the need for robust aggregation over multiple masks rather than single-shot estimation.

## 3.2 Exploiting Sparse Membership Signals in DLMs

Traditional LLM MIAs (Fu et al., 2024; Xie et al., 2024; Wang et al., 2025; Duan et al., 2023) show limited effectiveness when applied to DLMs, as demonstrated empirically in Section 4.2. These methods naturally follow the training objective in Equation (1) by approximating the expected loss difference through Monte Carlo estimation:

$$\Delta_{\mathrm{DF}}^{\mathrm{avg}}(\mathbf{x}) = \mathbb{E}_{\mathcal{S} \sim P(\mathcal{S})}[\Delta_{\mathrm{DF}}(\mathbf{x}; \mathcal{S})], \tag{4}$$

where $P(\mathcal{S})$ is the distribution over mask configurations used during training, and $\Delta_{\mathrm{DF}}(\mathbf{x}; \mathcal{S})$ is the loss difference for a specific mask configuration as defined in Equation (3). In practice, this expectation is estimated by randomly sampling multiple mask configurations and averaging the resulting loss differences.

However, this averaging-based approach is fundamentally limited in identifying membership signals for two key reasons. First, within a single configuration, averaging over all masked tokens includes noisy predictions that obscure genuine memorization—not all masked tokens contribute meaningful signal, as the strongest loss reductions often stem from domain adaptation effects on domain-specific tokens rather than instance-level memorization. Second, across configurations, random sampling with simple averaging treats all masking densities equally, ignoring that different densities operate at distinct signal-to-noise regimes: sparse masks provide stronger per-token signals with fewer aggregation points, while dense masks offer weaker individual signals but more aggregation opportunities.

This motivates us to address both issues through a different aggregation strategy. Within each configuration, instead of averaging over all masked tokens, we aggregate the sign of loss differences from multiple token subsets, as this binary decision is robust to outlier tokens that contribute primarily to noise. Across configurations, extracting meaningful membership information requires careful consideration of how these signals vary with masking density. Different masking densities provide different signal-to-noise characteristics, which is analogous to how different noise levels in image diffusion models capture different aspects of the data (Ho et al., 2020; Song et al., 2020). We exploit this by progressively increasing the masking density, gathering evidence at multiple scales rather than attempting to explore all possible configurations.

Our method comprises three key components: (1) **Robust subset aggregation**: At each density level, we sample multiple token subsets and employ sign-based statistics to aggregate their signals, ensuring robustness against noise; (2) **Progressive masking**: We systematically increase masking density to capture signals at multiple scales, from sparse masks (strong per-token signals but fewer aggregation points) to dense masks (weaker individual signals but more aggregation opportunities); and (3) **Adaptive weighting**: We weight contributions from different density levels, prioritizing the cleaner signals from sparser masks while incorporating cumulative evidence from denser configurations when beneficial.

## 3.3 Robust Subset Aggregation

At each progressive masking step $t \in \{1, \ldots, T\}$ with mask configuration $\mathcal{S}_t \subseteq [L]$ containing $k_t$ masked positions, we obtain losses for all masked tokens through a single forward pass. However, directly averaging all $k_t$ token losses is vulnerable to domination by extreme values, as a single token with exceptionally high or low loss can skew the entire signal. As we empirically demonstrate in Section E, the distribution of loss differences is governed by heavy-tailed statistics rather than Gaussian noise. Fine-tuning introduces domain adaptation effects that manifest as this long-tailed noise with occasional extreme values orders of magnitude larger than typical signals. These high-magnitude outliers often correspond to domain-specific tokens rather than instance-specific memorization. Consequently, a simple average is often overwhelmed by domain noise, whereas the true membership signal is consistent but sparse. To address this dual challenge, we employ a two-part robust aggregation strategy: local subset sampling combined with sign-based aggregation.

**Local Subset Sampling.** We sample $N$ random subsets of positions from $\mathcal{S}_t$, where each subset $\mathcal{U}^n \subset \mathcal{S}_t$, for $n = 1, \ldots, N$, contains $m$ positions (typically $m = 10$) with $m \ll k_t$. For each subset $\mathcal{U}^n$, we compute a localized loss difference:

$$\Delta_{\mathrm{DF}}^n(\mathbf{x}; \mathcal{S}_t) = \frac{1}{m} \sum_{i \in \mathcal{U}^n} [\ell_i^{\mathrm{R}}(\mathbf{x}, \mathcal{S}_t) - \ell_i^{\mathrm{T}}(\mathbf{x}, \mathcal{S}_t)], \tag{5}$$

where $i$ indexes positions in the sequence, and $\ell_i^{\mathrm{R}}(\mathbf{x}, \mathcal{S}_t)$, $\ell_i^{\mathrm{T}}(\mathbf{x}, \mathcal{S}_t)$ are the reference and target model losses at position $i$ when sequence $\mathbf{x}$ is masked according to $\mathcal{S}_t$.

This gives us $N$ different local measurements rather than a single global aggregate, each less susceptible to individual outlier tokens. This subsampling approach aligns with robust statistical methods where using multiple small random subsets provides stability against outliers (Rousseeuw & Leroy, 2003; Maronna et al., 2019). The principle is also employed in ensemble methods and stochastic optimization, where aggregating over random subsets yields more robust estimates than using all data points (Breiman, 1996; Bottou et al., 2018). Each subset excludes different tokens, so an extreme value that dominates one subset's signal appears in only a fraction of our $N$ measurements, while consistent membership signals present across most subsets accumulate through aggregation.

**Sign-Based Aggregation.** Rather than using the magnitude of each localized loss difference, we transform each $\Delta_{\mathrm{DF}}^n$ into a binary indicator: $B^n(\mathbf{x}) = \mathbf{1}[\Delta_{\mathrm{DF}}^n(\mathbf{x}; \mathcal{S}_t) > 0]$. This transformation discards magnitude information, recording only whether the reference model has a higher loss than the target model for that particular subset $\mathcal{U}^n$. While this might seem like discarding information, it actually provides robustness against heavy-tailed noise distributions.

For non-members, the target and reference models behave similarly since neither was trained on the sample. The loss difference $\Delta_{\mathrm{DF}}^n$ from subset $\mathcal{U}^n$ is purely noise—sometimes positive, sometimes negative, but centered at zero. Therefore, $B^n(\mathbf{x})$ equals 1 with probability exactly 0.5, regardless of the noise distribution's properties. This holds even if the noise has infinite variance or no defined mean, a robustness property that magnitude-based approaches cannot achieve.

For members, when we hit a mask configuration that activates memorization, the target model's loss drops below the reference model's loss, making $\Delta_{\mathrm{DF}}^n$ positive. Not every configuration activates memorization; most don't, but those that do consistently push $B^n$ toward 1. By aggregating these binary indicators across our $N$ sampled subsets, we compute $\hat{\beta}_t(\mathbf{x}) = \frac{1}{N} \sum_{n=1}^{N} B^n(\mathbf{x})$.

This principle that sign-based statistics remain efficient under minimal distributional assumptions is formalized in the Hodges-Lehmann theorem (Hodges Jr & Lehmann, 2011) and has been extensively validated in robust statistics literature (Huber, 1981; Peizer & Pratt, 1968). By combining local sampling with sign-based aggregation, we transform a single high-variance measurement into multiple robust binary decisions. This ensures that our membership detection at each masking density is stable and reliable, accumulating evidence from the consistency of the signal rather than its magnitude, which is precisely what we need for DLMs where sparse activation means most configurations contribute noise, but the few that reveal membership do so reliably in their direction rather than their size.

## 3.4 Progressive Masking with Adaptive Weighting

Given the step-wise scores $\{\hat{\beta}_t(\mathbf{x})\}_{t=1}^{T}$ from local sampling and sign-based aggregation, we now address how to effectively combine signals across different masking densities. As demonstrated

empirically in Figure 2, the membership signal $\Delta_{DF}(x; \mathcal{S})$ exhibits high variance across different configurations. Relying on a single masking density or a static configuration creates a high risk of sampling a "null" configuration where the true membership signal is drowned by noise. To mitigate this volatility, rather than fixing a single masking level, we progressively increase the masking density across $T$ steps. Specifically, at each step $t \in \{1, \ldots, T\}$, we set the masking density as:

$$\alpha_t = \alpha_{\min} + \frac{t-1}{T-1}(\alpha_{\max} - \alpha_{\min}), \tag{6}$$

where $\alpha_{\min}$ and $\alpha_{\max}$ (typically 5% to 50%) define the range of masking densities. This yields $k_t = \lceil L \cdot \alpha_t \rceil$ masked positions at step $t$.

This multi-scale approach exploits a fundamental trade-off in masking density. With sparse masks (low $\alpha_t$), each prediction benefits from rich surrounding context, making memorization patterns more apparent when they exist—but we have fewer masked positions to aggregate. With dense masks (high $\alpha_t$), we gain more aggregation points, but each individual prediction becomes noisier as less context is available and domain adaptation effects become more pronounced. Intuitively, this principle mirrors established strategies in statistical hypothesis testing. Similar to how multi-scale testing improves detection power by examining data at various resolutions to find sparse signals (Arias-Castro et al., 2011), our progressive masking captures memorization patterns that manifest differently across masking densities. Some memorization artifacts may only be detectable with sparse masks where strong contextual clues remain, while other patterns require aggregating evidence from many masked positions to overcome noise.

Empirical observation shows that signal quality degrades with masking density—early steps with sparse masks provide cleaner signals than later steps with dense masks. This motivates our adaptive weighting scheme:

$$\text{SAMA}(\mathbf{x}) = \sum_{t=1}^{T} w_t \hat{\beta}_t(\mathbf{x}) = \sum_{t=1}^{T} w_t \cdot \frac{1}{N} \sum_{n=1}^{N} \mathbf{1}[\Delta_{\text{DF}}^n(\mathbf{x}; \mathcal{S}_t) > 0], \tag{7}$$

where $w_t = \frac{1/t}{\sum_{i=1}^{T} 1/i}$ is the inverse-step weight inspired by the use of harmonic means in robust statistics (Hedges & Olkin, 2014), $\mathcal{S}_t$ denotes the mask configuration at step $t$, and $\Delta_{\text{DF}}^n(\mathbf{x}; \mathcal{S}_t)$ is the localized loss difference from Equation (5). This weighting prioritizes early steps where memorization signals are less contaminated by noise, while still incorporating evidence from all scales.

## 3.5 Algorithmic Specification

We now present the complete SAMA algorithm that implements our three-component strategy: progressive masking, robust subset aggregation, and adaptive weighting. The SAMA algorithm operates in two distinct phases to extract robust membership signals from DLMs. We theoretically and empirically demonstrate in Appendix D.7 that this robust aggregation logic incurs negligible computational overhead compared to standard query-based baselines.

**Phase I: Progressive Evidence Collection** (Algorithm 1). This phase implements our progressive masking strategy, systematically increasing the masking density across $T$ steps from $\alpha_{\min}$ to $\alpha_{\max}$ (typically 5% to 50% of tokens). At each step $t$, we determine the target number of masked positions $k_t$ and collect $N$ independent measurements through subset sampling uniformly without replacement.

**Phase II: Sign-based Aggregation and Weighting** (Algorithm 2). This phase transforms the raw loss differences into a robust membership score. First, each difference $\Delta$ is converted to a binary indicator $B = \mathbf{1}[\Delta > 0]$, implementing our sign-based detection that remains robust to heavy-tailed noise. For each step $t$, we compute the average binary indicator $\hat{\beta}_t$ across all $N$ samples at that density level.

---

**Algorithm 1:** Phase I: Progressive Evidence Collection

1 **Input:** Target model $\mathcal{M}_{\text{DF}}^{\text{T}}$, Reference model $\mathcal{M}_{\text{DF}}^{\text{R}}$, sequence $\mathbf{x}$ of length $L$
2 **Params:** Masking range $[\alpha_{\min}, \alpha_{\max}]$, steps $T$, samples $N$, subset size $m$
3 $\mathcal{D} \leftarrow \emptyset$
4 **for** $t = 1, \ldots, T$ **do**
    // Compute target masking density for step $t$
5    $k_t \leftarrow \lceil L \cdot (\alpha_{\min} + \frac{t-1}{T-1}(\alpha_{\max} - \alpha_{\min})) \rceil$
6    $\mathcal{D}_t \leftarrow []$
    // Sample and mask positions for this step
7    $\mathcal{S}_t \leftarrow \text{Sample}([L], k_t)$
8    $\mathbf{x}_{\text{m}} \leftarrow \text{Mask}(\mathbf{x}, \mathcal{S}_t)$
    // Compute losses
9    $\boldsymbol{\ell}^{\text{T}} \leftarrow \{-\log p_{\mathcal{M}_{\text{DF}}^{\text{T}}}(x_i|\mathbf{x}_{-\mathcal{S}_t})\}_{i \in \mathcal{S}_t}$
10    $\boldsymbol{\ell}^{\text{R}} \leftarrow \{-\log p_{\mathcal{M}_{\text{DF}}^{\text{R}}}(x_i|\mathbf{x}_{-\mathcal{S}_t})\}_{i \in \mathcal{S}_t}$
    // Sample $N$ local subsets from masked positions
11    **for** $n = 1, \ldots, N$ **do**
12        $\mathcal{U}^n \leftarrow \text{Sample}(\mathcal{S}_t, m)$
        // Compute subset difference
13        $\Delta^n \leftarrow \frac{1}{m} \sum_{i \in \mathcal{U}^n}(\ell_i^{\text{R}} - \ell_i^{\text{T}})$
14        $\mathcal{D}_t.\text{append}(\Delta^n)$
15    $\mathcal{D}.\text{add}((t, \mathcal{D}_t))$
16 **Return:** difference collection $\mathcal{D}$

---

**Algorithm 2:** Phase II: Binary Aggregation & Weighting

1 **Input:** Difference collection $\mathcal{D} = \{(t, \mathcal{D}_t)\}_{t=1}^T$
2 $\mathcal{B} \leftarrow []$     // Stepwise sign stats
3 **foreach** $(t, \mathcal{D}_t) \in \mathcal{D}$ **do**
4    $\mathcal{B}_t \leftarrow []$
5    **foreach** $\Delta^n \in \mathcal{D}_t$ **do**
6        $\mathcal{B}_t.\text{append}(\mathbf{1}[\Delta^n > 0])$
7    $\hat{\beta}_t \leftarrow \text{Mean}(\mathcal{B}_t)$
8    $\mathcal{B}.\text{append}((t, \hat{\beta}_t))$
    // Compute inverse-step weights
9 $H \leftarrow \sum_{i=1}^T 1/i$
10 $\mathbf{w} \leftarrow [(1/t)/H \text{ for } t = 1, \ldots, T]$   // Eq. 7
    // Final weighted aggregation
11 $\phi \leftarrow \sum_{t=1}^T w_t \cdot \hat{\beta}_t$
12 **Return:** membership score $\phi \in [0, 1]$

---

**Algorithm 3:** SAMA: Main Procedure

1 **Input:** Target model $\mathcal{M}_{\text{DF}}^{\text{T}}$, Reference model $\mathcal{M}_{\text{DF}}^{\text{R}}$, Text sequence $\mathbf{x}$
2 **Params:** Masking range $[\alpha_{\min}, \alpha_{\max}]$, Steps $T$, Samples $N$, Subset size $m$
3 $\mathcal{D} \leftarrow$ Algorithm 1$(\mathcal{M}_{\text{DF}}^{\text{T}}, \mathcal{M}_{\text{DF}}^{\text{R}}, \mathbf{x}, \alpha_{\min}, \alpha_{\max}, T, N, m)$
4 $\phi \leftarrow$ Algorithm 2$(\mathcal{D})$
5 **Return:** membership score $\phi$

---

# 4 Experiments

This section validates our proposed SAMA attack against state-of-the-art DLMs, demonstrating its effectiveness across diverse datasets and comparing it to existing MIA methods.

## 4.1 Experimental Setup

We evaluate SAMA on two state-of-the-art diffusion language models: LLaDA-8B-Base (Nie et al., 2025) and Dream-v0-7B-Base (Ye et al., 2025a), with their pre-trained versions serving as reference models $\mathcal{M}_{\text{DF}}^{\text{R}}$. Fine-tuning was performed on member datasets from six diverse domains of the MIMIR benchmark (Duan et al., 2024) following Zhang et al. (2025b); Chang et al. (2024); Antebi et al. (2025), and three standard NLP benchmarks (WikiText-103 (Merity et al., 2016), AG News (Zhang et al., 2015), XSum (Narayan et al., 2018)) following Fu et al. (2024). The fine-tuned models maintain strong utility (see Section D.2).

SAMA uses $T = 16$ progressive masking steps from $\alpha_{\min} = 5\%$ to $\alpha_{\max} = 50\%$, sampling $N = 128$ subsets of size $m = 10$ tokens at each step. We compare against twelve baselines: (1) Autoregressive MIA methods: Loss (Yeom et al., 2018), ZLIB (Carlini et al., 2021), Lowercase (Carlini et al., 2021), Neighbor (Mattern et al., 2023), Min-K% (Shi et al., 2024b), Min-K%++ (Zhang et al., 2025a), ReCall (Xie et al., 2024), CON-ReCall (Wang et al., 2025), BoWs (Das et al., 2025), and Ratio (Watson et al., 2021); (2) Diffusion MIA methods adapted from image domain: SecMI (Duan et al., 2023) and PIA (Kong et al., 2023). Evaluation follows standard MIA metrics (Carlini et al., 2022): AUC and TPR at 10%, 1%, and 0.1% FPR. All baselines also utilize an identical budget of $T = 16$ model queries per sample. Full experimental details are provided in Section D.1.

## 4.2 Main Results

Table 1 presents the MIA performance of SAMA compared to twelve baseline methods across six diverse datasets from MIMIR, evaluated using AUC and TPR at various FPR thresholds (additional results on Dream-v0-7B and three NLP benchmarks in Section D.3).

Table 1: MIA performance (AUC, TPR@10%FPR, TPR@1%FPR, TPR@0.1%FPR) across datasets. Each cell shows the mean with std. dev. as a subscript. The best results are highlighted.

| MIAs | ArXiv | | | | GitHub | | | | HackerNews | | | |
|---|---|---|---|---|---|---|---|---|---|---|---|---|
| | AUC | T@10% | T@1% | T@0.1% | AUC | T@10% | T@1% | T@0.1% | AUC | T@10% | T@1% | T@0.1% |
| Loss | $0.506_{\pm.01}$ | $0.119_{\pm.02}$ | $0.010_{\pm.01}$ | $0.000_{\pm.00}$ | $0.551_{\pm.01}$ | $0.161_{\pm.02}$ | $0.036_{\pm.01}$ | $0.007_{\pm.01}$ | $0.495_{\pm.01}$ | $0.118_{\pm.01}$ | $0.010_{\pm.01}$ | $0.000_{\pm.00}$ |
| ZLIB | $0.490_{\pm.01}$ | $0.119_{\pm.02}$ | $0.012_{\pm.00}$ | $0.000_{\pm.00}$ | $0.561_{\pm.01}$ | $0.205_{\pm.02}$ | $0.045_{\pm.01}$ | $0.007_{\pm.00}$ | $0.486_{\pm.01}$ | $0.083_{\pm.01}$ | $0.009_{\pm.01}$ | $0.001_{\pm.00}$ |
| Lowercase | $0.515_{\pm.01}$ | $0.103_{\pm.01}$ | $0.013_{\pm.00}$ | $0.001_{\pm.00}$ | $0.579_{\pm.01}$ | $0.178_{\pm.02}$ | $0.044_{\pm.01}$ | $0.008_{\pm.01}$ | $0.483_{\pm.01}$ | $0.074_{\pm.01}$ | $0.007_{\pm.00}$ | $0.001_{\pm.00}$ |
| Min-K% | $0.488_{\pm.01}$ | $0.102_{\pm.01}$ | $0.012_{\pm.00}$ | $0.001_{\pm.00}$ | $0.530_{\pm.01}$ | $0.171_{\pm.02}$ | $0.039_{\pm.01}$ | $0.007_{\pm.01}$ | $0.492_{\pm.01}$ | $0.108_{\pm.01}$ | $0.013_{\pm.01}$ | $0.001_{\pm.00}$ |
| Min-K%++ | $0.485_{\pm.01}$ | $0.095_{\pm.01}$ | $0.006_{\pm.00}$ | $0.000_{\pm.00}$ | $0.496_{\pm.01}$ | $0.117_{\pm.01}$ | $0.016_{\pm.01}$ | $0.003_{\pm.00}$ | $0.486_{\pm.01}$ | $0.100_{\pm.02}$ | $0.008_{\pm.00}$ | $0.002_{\pm.00}$ |
| BoWs | $0.519_{\pm.01}$ | $0.107_{\pm.01}$ | $0.011_{\pm.00}$ | $0.000_{\pm.00}$ | $0.656_{\pm.02}$ | $0.306_{\pm.02}$ | $0.154_{\pm.02}$ | $0.059_{\pm.05}$ | $0.527_{\pm.01}$ | $0.128_{\pm.01}$ | $0.009_{\pm.01}$ | $0.001_{\pm.00}$ |
| ReCall | $0.501_{\pm.01}$ | $0.132_{\pm.02}$ | $0.007_{\pm.00}$ | $0.000_{\pm.00}$ | $0.562_{\pm.01}$ | $0.187_{\pm.01}$ | $0.037_{\pm.01}$ | $0.007_{\pm.01}$ | $0.494_{\pm.01}$ | $0.090_{\pm.01}$ | $0.010_{\pm.01}$ | $0.000_{\pm.00}$ |
| CON-Recall | $0.500_{\pm.02}$ | $0.101_{\pm.02}$ | $0.011_{\pm.00}$ | $0.000_{\pm.00}$ | $0.549_{\pm.01}$ | $0.168_{\pm.01}$ | $0.027_{\pm.01}$ | $0.005_{\pm.00}$ | $0.501_{\pm.02}$ | $0.098_{\pm.01}$ | $0.015_{\pm.01}$ | $0.000_{\pm.00}$ |
| Neighbor | $0.506_{\pm.01}$ | $0.098_{\pm.01}$ | $0.012_{\pm.01}$ | $0.001_{\pm.00}$ | $0.478_{\pm.01}$ | $0.072_{\pm.01}$ | $0.008_{\pm.01}$ | $0.000_{\pm.00}$ | $0.520_{\pm.01}$ | $0.123_{\pm.01}$ | $0.009_{\pm.00}$ | $0.001_{\pm.00}$ |
| Ratio | $0.597_{\pm.01}$ | $0.181_{\pm.01}$ | $0.023_{\pm.01}$ | $0.001_{\pm.00}$ | $0.743_{\pm.01}$ | $0.355_{\pm.03}$ | $0.081_{\pm.02}$ | $0.017_{\pm.01}$ | $0.575_{\pm.02}$ | $0.146_{\pm.01}$ | $0.013_{\pm.01}$ | $0.000_{\pm.00}$ |
| SecMI | $0.520_{\pm.01}$ | $0.096_{\pm.02}$ | $0.006_{\pm.00}$ | $0.001_{\pm.00}$ | $0.604_{\pm.01}$ | $0.190_{\pm.01}$ | $0.044_{\pm.01}$ | $0.019_{\pm.01}$ | $0.523_{\pm.02}$ | $0.125_{\pm.02}$ | $0.013_{\pm.00}$ | $0.001_{\pm.00}$ |
| PIA | $0.525_{\pm.01}$ | $0.099_{\pm.01}$ | $0.011_{\pm.01}$ | $0.000_{\pm.00}$ | $0.571_{\pm.01}$ | $0.131_{\pm.01}$ | $0.012_{\pm.00}$ | $0.001_{\pm.00}$ | $0.494_{\pm.01}$ | $0.086_{\pm.01}$ | $0.014_{\pm.00}$ | $0.000_{\pm.00}$ |
| **SAMA (Ours)** | $\mathbf{0.850}_{\pm.01}$ | $\mathbf{0.586}_{\pm.03}$ | $\mathbf{0.178}_{\pm.03}$ | $\mathbf{0.014}_{\pm.01}$ | $\mathbf{0.876}_{\pm.01}$ | $\mathbf{0.647}_{\pm.03}$ | $\mathbf{0.259}_{\pm.05}$ | $\mathbf{0.075}_{\pm.05}$ | $\mathbf{0.657}_{\pm.01}$ | $\mathbf{0.215}_{\pm.02}$ | $\mathbf{0.027}_{\pm.01}$ | $\mathbf{0.003}_{\pm.00}$ |

| MIAs | PubMed Central | | | | Wikipedia (en) | | | | Pile CC | | | |
|---|---|---|---|---|---|---|---|---|---|---|---|---|
| | AUC | T@10% | T@1% | T@0.1% | AUC | T@10% | T@1% | T@0.1% | AUC | T@10% | T@1% | T@0.1% |
| Loss | $0.498_{\pm.01}$ | $0.114_{\pm.02}$ | $0.016_{\pm.01}$ | $0.001_{\pm.00}$ | $0.495_{\pm.01}$ | $0.087_{\pm.00}$ | $0.010_{\pm.00}$ | $0.003_{\pm.00}$ | $0.502_{\pm.01}$ | $0.105_{\pm.01}$ | $0.012_{\pm.00}$ | $0.002_{\pm.00}$ |
| ZLIB | $0.488_{\pm.01}$ | $0.096_{\pm.02}$ | $0.005_{\pm.00}$ | $0.001_{\pm.00}$ | $0.495_{\pm.01}$ | $0.093_{\pm.01}$ | $0.008_{\pm.00}$ | $0.000_{\pm.00}$ | $0.491_{\pm.01}$ | $0.095_{\pm.01}$ | $0.007_{\pm.00}$ | $0.001_{\pm.00}$ |
| Lowercase | $0.502_{\pm.01}$ | $0.096_{\pm.01}$ | $0.002_{\pm.00}$ | $0.000_{\pm.00}$ | $0.535_{\pm.01}$ | $0.107_{\pm.02}$ | $0.013_{\pm.00}$ | $0.001_{\pm.00}$ | $0.518_{\pm.01}$ | $0.102_{\pm.01}$ | $0.009_{\pm.00}$ | $0.001_{\pm.00}$ |
| Min-K% | $0.500_{\pm.01}$ | $0.119_{\pm.01}$ | $0.008_{\pm.00}$ | $0.002_{\pm.00}$ | $0.482_{\pm.01}$ | $0.070_{\pm.01}$ | $0.008_{\pm.00}$ | $0.000_{\pm.00}$ | $0.491_{\pm.01}$ | $0.095_{\pm.01}$ | $0.008_{\pm.00}$ | $0.001_{\pm.00}$ |
| Min-K%++ | $0.494_{\pm.01}$ | $0.109_{\pm.01}$ | $0.011_{\pm.00}$ | $0.000_{\pm.00}$ | $0.488_{\pm.01}$ | $0.118_{\pm.02}$ | $0.004_{\pm.00}$ | $0.000_{\pm.00}$ | $0.491_{\pm.01}$ | $0.113_{\pm.01}$ | $0.007_{\pm.00}$ | $0.001_{\pm.00}$ |
| BoWs | $0.489_{\pm.01}$ | $0.100_{\pm.01}$ | $0.002_{\pm.00}$ | $0.000_{\pm.00}$ | $0.471_{\pm.01}$ | $0.084_{\pm.02}$ | $0.003_{\pm.00}$ | $0.001_{\pm.00}$ | $0.480_{\pm.01}$ | $0.092_{\pm.01}$ | $0.003_{\pm.00}$ | $0.001_{\pm.00}$ |
| ReCall | $0.495_{\pm.01}$ | $0.088_{\pm.01}$ | $0.007_{\pm.00}$ | $0.000_{\pm.00}$ | $0.506_{\pm.01}$ | $0.103_{\pm.01}$ | $0.010_{\pm.01}$ | $0.000_{\pm.00}$ | $0.500_{\pm.01}$ | $0.096_{\pm.01}$ | $0.009_{\pm.00}$ | $0.000_{\pm.00}$ |
| CON-Recall | $0.498_{\pm.02}$ | $0.119_{\pm.02}$ | $0.009_{\pm.00}$ | $0.000_{\pm.00}$ | $0.498_{\pm.02}$ | $0.092_{\pm.01}$ | $0.008_{\pm.00}$ | $0.000_{\pm.00}$ | $0.498_{\pm.01}$ | $0.106_{\pm.01}$ | $0.009_{\pm.00}$ | $0.000_{\pm.00}$ |
| Neighbor | $0.506_{\pm.01}$ | $0.091_{\pm.01}$ | $0.011_{\pm.01}$ | $0.000_{\pm.00}$ | $0.504_{\pm.01}$ | $0.101_{\pm.01}$ | $0.009_{\pm.00}$ | $0.001_{\pm.00}$ | $0.505_{\pm.01}$ | $0.096_{\pm.01}$ | $0.010_{\pm.00}$ | $0.001_{\pm.00}$ |
| Ratio | $0.555_{\pm.01}$ | $0.128_{\pm.02}$ | $0.018_{\pm.01}$ | $0.003_{\pm.01}$ | $0.653_{\pm.01}$ | $0.184_{\pm.03}$ | $0.011_{\pm.01}$ | $0.000_{\pm.00}$ | $0.604_{\pm.01}$ | $0.156_{\pm.02}$ | $0.015_{\pm.01}$ | $0.002_{\pm.00}$ |
| SecMI | $0.510_{\pm.01}$ | $0.109_{\pm.01}$ | $0.009_{\pm.00}$ | $0.008_{\pm.00}$ | $0.522_{\pm.01}$ | $0.101_{\pm.01}$ | $0.015_{\pm.00}$ | $0.009_{\pm.00}$ | $0.515_{\pm.01}$ | $0.108_{\pm.01}$ | $0.010_{\pm.00}$ | $0.001_{\pm.00}$ |
| PIA | $0.496_{\pm.01}$ | $0.102_{\pm.01}$ | $0.005_{\pm.00}$ | $0.000_{\pm.00}$ | $0.522_{\pm.01}$ | $0.120_{\pm.01}$ | $0.011_{\pm.00}$ | $0.001_{\pm.00}$ | $0.509_{\pm.01}$ | $0.103_{\pm.01}$ | $0.009_{\pm.00}$ | $0.000_{\pm.00}$ |
| **SAMA (Ours)** | $\mathbf{0.814}_{\pm.01}$ | $\mathbf{0.442}_{\pm.03}$ | $\mathbf{0.132}_{\pm.03}$ | $\mathbf{0.011}_{\pm.01}$ | $\mathbf{0.790}_{\pm.01}$ | $\mathbf{0.433}_{\pm.03}$ | $\mathbf{0.136}_{\pm.01}$ | $\mathbf{0.008}_{\pm.02}$ | $\mathbf{0.778}_{\pm.01}$ | $\mathbf{0.408}_{\pm.03}$ | $\mathbf{0.115}_{\pm.02}$ | $\mathbf{0.009}_{\pm.01}$ |

SAMA demonstrates substantial and consistent improvements across all metrics and datasets. On average, SAMA achieves an AUC of 0.81, compared to the best baseline (Ratio) at 0.62—a 30% relative improvement. The superiority is particularly pronounced at low FPR thresholds, critical for practical deployment. For instance, at TPR@1%FPR, SAMA achieves 0.16 on average while the best baseline reaches only 0.04, representing a 4× improvement. On the GitHub dataset, where memorization is most pronounced, SAMA attains an AUC of 0.88 with TPR@10%FPR of 0.65, while the best baseline (Ratio) achieves 0.74 and 0.36, respectively.

Notably, most traditional MIA methods designed for autoregressive models perform near randomly (AUC $\approx$ 0.50) on DLMs, confirming that existing approaches fail to capture the unique memorization patterns in diffusion-based architectures. Even methods specifically designed for the diffusion paradigm (SecMI, PIA) show limited effectiveness, with AUCs barely exceeding 0.52. This validates our hypothesis that DLMs require fundamentally different attack strategies that account for their sparse configuration-dependent membership signals. We also evaluate SAMA's effectiveness against privacy-preserving fine-tuning methods (Differential Privacy Liu et al. (2024) and LoRA Hu et al. (2021)) in Section D.5.

## 4.3 Ablation Study

Figure 3 shows the contribution of each SAMA component. Starting from a baseline loss attack (AUC $\approx$ 0.5), we progressively add: (1) reference model calibration, yielding 0.09-0.19 AUC improvement by isolating fine-tuning-specific memorization; (2) progressive masking, contributing modest 2-3% gains by capturing multi-scale patterns; (3) robust subset aggregation, delivering the

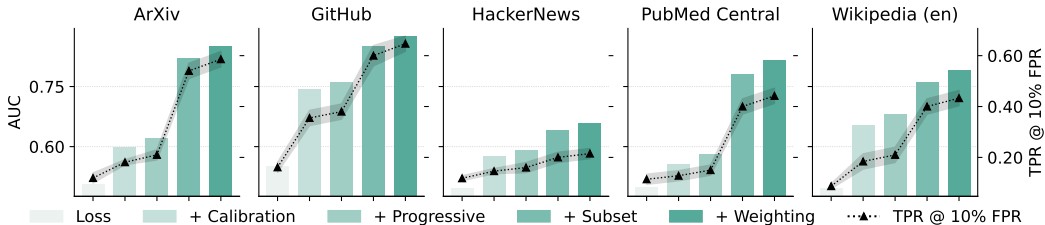

Figure 3: (Ablation) Impact of SAMA's Core Components Across MIMIR Datasets. The bar charts are illustrated in AUC. The overlaid line plots show the corresponding TPR@10%FPR.

largest improvement (20-30% AUC increase) by robustly handling heavy-tailed noise through sign-based voting rather than magnitude averaging; and (4) adaptive weighting, providing 3-5% final refinement by prioritizing cleaner signals from sparse masks. The consistent pattern across datasets confirms that sign-based aggregation is the critical component for handling DLMs' sparse, noisy signals, with calibration and progressive strategies providing essential but smaller contributions. We provide an extended sensitivity analysis of hyperparameters (steps $T$, subset size $m$, number of subsets $N$) in Section D.4.

# 5 Related Work

Our work on SAMA builds upon research in three primary areas: MIA on ARMs, MIA for image diffusion models, and the evolution of DLMs. For an expanded discussion, please refer to Section C.

**MIAs on ARM and Diffusion-based Generative Models.** Membership Inference Attacks (MIAs) (Shokri et al., 2017) have demonstrated particular vulnerabilities in fine-tuned Autoregressive Models (ARMs) (Mireshghallah et al., 2022b; Fu et al., 2024; Zeng et al., 2024; Puerto et al., 2025). Standard MIA techniques often rely on loss analysis or reference model calibration (Watson et al., 2021; Mireshghallah et al., 2022a; Fu et al., 2024), primarily targeting signals from the unidirectional, final-layer outputs of ARMs. This fundamentally differs from SAMA's approach to bidirectional diffusion models. Prior MIA research for diffusion models has predominantly focused on *image* generation, identifying vulnerabilities related to loss, gradients, and intermediate denoising steps (Hu & Pang, 2023; Pang et al., 2024; Matsumoto et al., 2023; Kong et al., 2023; Duan et al., 2023; Fu et al., 2023; Zhai et al., 2024). However, methods designed for pre-trained models often fail on fine-tuned models as they primarily capture domain adaptation signals rather than instance-specific memorization—a challenge that SAMA addresses through robust aggregation designed to isolate true membership signals.

**Diffusion Language Models.** The adaptation of diffusion principles (Sohl-Dickstein et al., 2015; Ho et al., 2020) to discrete language has led to Masked Diffusion Models (MDMs) (Austin et al., 2023). These models, exemplified by recent large-scale works like LLaDA (Nie et al., 2025) and Dream (Ye et al., 2025a), iteratively mask and predict tokens, establishing a viable alternative to ARMs. SAMA is specifically designed to exploit this core iterative mechanism.

# 6 Conclusions

This work presents the first systematic investigation of membership inference vulnerabilities in Diffusion Language Models. DLMs' bidirectional masking mechanism creates exploitable memorization patterns absent in autoregressive models. Our SAMA leverages this through robust subset aggregation and progressive masking, achieving over $10\times$ improvement compared to existing baselines. These findings reveal key privacy risks in emerging DLMs. *Limitations:* SAMA targets mask-predict diffusion models; its effectiveness on other diffusion paradigms remains unexplored. The attack requires reference models with compatible tokenizers and masking schemes (see Section D.6). While SAMA's progressive masking is specialized for the bidirectional nature of DLMs, the robust subset aggregation component is generalizable. The insight of using sign-based voting to filter heavy-tailed domain noise could potentially enhance MIAs against autoregressive or masked language models in future work.

## Ethics and Broader Impact Statements

Our work introduces SAMA, a potent membership inference attack that exposes privacy vulnerabilities in diffusion-based LLMs by identifying training data. While this research highlights critical risks, all experiments were conducted strictly on public datasets (e.g., MIMIR (Duan et al., 2024), WikiText-103 (Merity et al., 2016), AG News (Zhang et al., 2015), and XSum (Narayan et al., 2018)) to prevent any new privacy violations. By demonstrating SAMA's effectiveness, this paper reveals previously overlooked vulnerabilities specific to diffusion-based LLMs. This research directly motivates and informs the development of tailored privacy-preserving defenses for this emerging LLM class, contributing to safer AI model deployment.

## Reproducibility Statement

To ensure reproducibility of our results, we provide comprehensive implementation details throughout the paper and supplementary materials. The complete SAMA algorithm is formally specified in Section 3.5 with all hyperparameters explicitly stated in Section 4.1. Full experimental details, including fine-tuning procedures, model configurations, and evaluation protocols, are provided in Section D.1. We use publicly available models (LLaDA-8B-Base and Dream-v0-7B-Base) and standard benchmarks (MIMIR, WikiText-103, AG News, XSum) with data processing steps detailed in Section D.1. The mathematical formulation and theoretical foundations are presented in Section 2 and Section 3.1. Implementation code for SAMA and all baseline methods, along with scripts for reproducing experiments, will be made available at `https://github.com/Stry233/SAMA`. Additional results validating our findings across different models and datasets are provided in Section D.3, demonstrating the robustness of our method.

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

# A  Notation Summary

This section provides a summary of core mathematical notation used throughout the paper to ensure clarity and consistency. Table 2 lists these symbols and their meanings in the context of SAMA.

# B  LLM Usage Disclosure

We used Large Language Models OpenAI (2024) to aid in polishing the writing of this paper. Specifically, LLMs were used to refine the abstract for clarity and conciseness, and to generate summary statements. All technical content, experimental design, analysis, and core contributions are entirely the authors' original work. The LLM served only as a writing assistant for improving readability and presentation of already-developed ideas.

# C  Additional Related Works

**MIA on ARMs.**  MIAs against ARMs initially showed limited success on pre-training data under rigorous IID evaluation (Duan et al., 2024; Meeus et al., 2024; Das et al., 2025), attributed to large datasets, few training epochs, and fuzzy text boundaries (Duan et al., 2024; Carlini et al., 2021; Satvaty et al., 2025). Early LLM MIAs adapted loss/perplexity thresholding (Yeom et al., 2018; Shokri et al., 2017) or reference model calibration (Watson et al., 2021; Mireshghallah et al., 2022a), with LiRA (Carlini et al., 2022) emphasizing low-FPR evaluation. Reference-free methods like Min-K% (Shi et al., 2024b) and the theoretically grounded Min-K%++ (Zhang et al., 2025a) analyzed token probabilities. Other approaches compare outputs on perturbed neighbors (Mattern et al., 2023), leverage semantic understanding (SMIA) (Mozaffari & Marathe, 2024), analyze loss trajectories (Liu et al., 2022; Li et al., 2024), or target memorization via probabilistic variation (SPV-MIA) (Fu et al., 2024). Fine-tuned LLMs proved more vulnerable (Mireshghallah et al., 2022b; Zeng et al., 2024), leading to attacks like SPV-MIA with self-prompt calibration (Fu et al., 2024) and User Inference targeting user participation (Kandpal et al., 2024). Extraction methods (Carlini et al., 2021) and label-only attacks (He et al., 2025; Wen et al., 2024) were also developed. Recent findings suggest MIA effectiveness against pre-trained LLMs increases significantly when aggregating weak signals over longer sequences (Puerto et al., 2025). However, nearly all existing LLM MIAs fundamentally probe signals derived from the final layer outputs or aggregated scores pertinent to the autoregressive generation process.

**MIA for Diffusion-Based Generative Models.**  Investigating MIAs in diffusion models initially centered on image generation, posing distinct challenges compared to GANs or VAEs (Duan et al., 2023; Carlini et al., 2021). White-box analyses confirmed vulnerability, identifying loss and likelihood as viable signals (Hu & Pang, 2023) and suggesting intermediate denoising timesteps carry heightened risk (Matsumoto et al., 2023), with gradients potentially offering stronger leakage channels (Pang et al., 2024). Despite this, mounting effective black-box attacks, especially against large-scale models like Stable Diffusion, proved difficult (Dubiński et al., 2023). Significant *grey-box* attacks emerged, leveraging access beyond final outputs. Reconstruction-based methods showed promise, particularly for fine-tuned models (Pang & Wang, 2023). Trajectory analysis, probing the iterative denoising path, yielded query-efficient attacks like PIA (Kong et al., 2023). Other black-box innovations bypassed shadow models by analyzing generated distributions (Zhang et al., 2024) or employing quantile regression (Tang et al., 2023). Advanced methods aim for deeper insights, detecting memorization via probabilistic fluctuations (PFAMI (Fu et al., 2023)) or optimizing attacks through noise analysis (OMS (Fu et al., 2025)). Text-to-image models introduced further complexities and attack surfaces (Wu et al., 2022), exploited via techniques like conditional overfitting analysis (CLiD (Zhai et al., 2024)). These studies highlight the importance of the iterative process in image diffusion privacy but are tailored to its continuous state space and noise-based corruption, distinct from the mechanisms in discrete language diffusion.

**Diffusion-Based Language Models.**  Adapting diffusion principles (Sohl-Dickstein et al., 2015; Ho et al., 2020; Song et al., 2020) to discrete language required overcoming unique hurdles. Ap-

Table 2: Summary of Core Notation.

| Symbol | Meaning |
|---|---|
| *General Symbols* | |
| $\mathbf{x} = \{x_i\}_{i=1}^{L}$ | A text sequence (data record). |
| $\mathcal{V}$ | Model's token vocabulary. |
| $L$ | Length of the sequence $\mathbf{x}$. |
| $[L]$ | The set $\{1, 2, ..., L\}$ of all token positions. |
| $x_i$ | The $i$-th token in sequence $\mathbf{x}$. |
| $x_{<i}$ | Tokens preceding position $i$: $\{x_1, ..., x_{i-1}\}$. |
| $D_{\text{train}}$ | Training dataset (member set). |
| $\mu(\mathbf{x})$ | True membership status of $\mathbf{x}$ (1 for member, 0 for non-member). |
| $\hat{\mu}(\mathbf{x})$ | Predicted membership status of $\mathbf{x}$. |
| $A(\mathbf{x}, \mathcal{M}_{\text{DF}}^{\text{T}})$ | Attack function outputting membership prediction. |
| *Model and Loss Notation* | |
| $\mathcal{M}_{\text{DF}}^{\text{T}}$ | Target diffusion language model (fine-tuned). |
| $\mathcal{M}_{\text{DF}}^{\text{R}}$ | Reference diffusion language model (pre-trained base). |
| $\mathcal{M}_{\text{AR}}^{\text{T}}$ | Target autoregressive model. |
| $\mathcal{M}_{\text{AR}}^{\text{R}}$ | Reference autoregressive model. |
| $\mathcal{S} \subseteq [L]$ | A mask configuration (set of masked positions). |
| $\mathbf{x}_{-\mathcal{S}}$ | Unmasked context: $\{x_i : i \in [L] \setminus \mathcal{S}\}$. |
| $\ell(\mathbf{x}, \mathcal{S})$ | Training loss for DLMs (Equation (1)). |
| $p(x_i \mid \mathbf{x}_{-\mathcal{S}})$ | Probability of token $x_i$ given unmasked context (DLMs). |
| $p(x_i \mid x_{<i})$ | Probability of token $x_i$ given preceding context (ARMs). |
| $\ell_i(\mathbf{x}, \mathcal{S})$ | Loss at position $i$ in DLMs: $-\log p(x_i \mid \mathbf{x}_{-\mathcal{S}})$. |
| $\ell_i^{\text{R}}(\mathbf{x}, \mathcal{S}), \ell_i^{\text{T}}(\mathbf{x}, \mathcal{S})$ | Per-token losses for reference and target models. |
| SAMA *Algorithm Symbols* | |
| $T$ | Number of progressive masking steps. |
| $t$ | Step index ($1 \leq t \leq T$). |
| $[\alpha_{\min}, \alpha_{\max}]$ | Range of masking densities (e.g., [0.05, 0.5]). |
| $\alpha_t$ | Masking density at step $t$. |
| $k_t$ | Number of masked positions at step $t$: $\lceil L \cdot \alpha_t \rceil$. |
| $N$ | Number of samples (random subsets) per step. |
| $m$ | Subset size (typically 10 tokens). |
| $\mathcal{S}_t$ | Mask configuration at step $t$. |
| $\mathcal{U}^n$ | The $n$-th sampled subset from $\mathcal{S}_t$. |
| $\Delta^n$ | Loss difference for $n$-th subset. |
| $B^n$ | Binary indicator: $\mathbf{1}[\Delta^n > 0]$. |
| $\hat{\beta}_t$ | Average binary indicator at step $t$. |
| $w_t$ | Weight for step $t$: $(1/t)/H$ where $H = \sum_{i=1}^{T} 1/i$. |
| $H$ | Harmonic sum for normalization: $\sum_{i=1}^{T} 1/i$. |
| SAMA$(\mathbf{x})$ | Final membership score (Equation (7)). |
| $\phi$ | Alternative notation for final membership score. |
| $\mathcal{D}$ | Collection of loss differences across all steps. |
| $\mathcal{D}_t$ | Loss differences collected at step $t$. |

proaches include diffusing continuous text embeddings (Li et al., 2022; Gong et al., 2022; Han et al., 2022; Strudel et al., 2022; Chen et al., 2022; Dieleman et al., 2022; Richemond et al., 2022; Wu et al., 2023; Mahabadi et al., 2023; Ye et al., 2023b) or modeling continuous proxies for discrete distributions (Lou & Ermon, 2023; Graves et al., 2023; Lin et al., 2023; Xue et al., 2024), though often encountering scalability challenges relative to ARMs (Gulrajani & Hashimoto, 2023). Discrete diffusion frameworks, operating directly on tokens (Austin et al., 2021; Hoogeboom et al., 2021b;a; He et al., 2022; Campbell et al., 2022; Meng et al., 2022; Reid et al., 2022; Sun et al., 2022; Kitouni et al., 2023; Zheng et al., 2023b; Chen et al., 2023; Ye et al., 2023a; Gat et al., 2024; Zheng et al., 2024), offered an alternative path. Among these, Masked Diffusion Models (MDMs) (Austin et al., 2021), utilizing iterative masking and Transformer-based prediction, showed early promise (Lou et al., 2023; Nie et al., 2024), supported by theoretical work (Ou et al., 2024; Shi et al., 2024a). Research focused on refining MDMs (Sahoo et al., 2024), integrating them with ARM pre-training (Gong et al., 2024), and acceleration (Kou et al., 2024; Xu et al., 2025). This progress culminated in models like LLaDA (Nie et al., 2025) and Dream (Ye et al., 2025a), demonstrating that large-scale discrete diffusion models can achieve emergent LLM capabilities comparable to strong ARMs, establishing them as a viable, distinct LLM paradigm.

Despite the rapid maturation of diffusion-based text generation, particularly with capable models like LLaDA and Dream, their vulnerability to MIAs remains largely uncharted. Existing MIA techniques are ill-suited: those for ARMs target likelihoods from sequential generation, while those for image diffusion address continuous denoising steps. The unique discrete "mask-and-predict" iterative mechanism of diffusion-based LLMs necessitates novel MIA formulations. Inspired by stepwise analysis concepts proven effective in the (continuous) image domain (Duan et al., 2023; Kong et al., 2023), our work tackles this challenge directly.

# D  Supplementary Experimental Results

This section provides supplementary materials to complement the main paper. It begins with a detailed breakdown of the experimental settings, including dataset specifics, model configurations, fine-tuning procedures, and comprehensive descriptions of baseline methods and evaluation metrics. Subsequently, we present extended results for our main comparisons and ablation studies across a wider range of conditions. Further analyses delve into the impact of model utility, training duration, and reference model choice on attack efficacy. Finally, we explore potential defense mechanisms against SAMA and offer a comparative perspective on the vulnerability of diffusion-based models versus their autoregressive counterparts.

## D.1  Experiment Settings

This section provides comprehensive details regarding the experimental setup used to evaluate SAMA, supplementing Section 4.1 of the main paper. We describe our evaluation across nine diverse datasets spanning scientific, code, and general text domains, with samples ranging from 1,000 to 10,000 per dataset. Our experiments focus on two state-of-the-art diffusion language models fine-tuned using consistent hyperparameters. We compare SAMA against twelve baseline MIA methods, including ten adapted from autoregressive models and two designed for diffusion models, with all methods using 16 Monte Carlo samples for fair comparison. Performance is assessed through standard MIA metrics (AUC, TPR at various FPR thresholds) and model utility measures (perplexity, LLM-as-a-Judge evaluation).

### D.1.1  Datasets

Our experiments utilize datasets from two sources, summarized in Table 3. The primary evaluation uses six subsets from the **MIMIR benchmark** (Duan et al., 2024), following the `13_0.8` deduplication protocol from (Wang et al., 2025). Additionally, we evaluate on three standard NLP tasks following Fu et al. (2024): WikiText-103 (Merity et al., 2016), AG News (Zhang et al., 2015), XSum (Narayan et al., 2018).

Table 3: Dataset overview for MIA evaluation.

| Dataset | Type | Train | Test |
|---|---|---|---|
| *MIMIR Benchmark (`13_0.8 split`)* | | | |
| ArXiv | Scientific | 1,000 | 1,000 |
| GitHub | Code | 1,000 | 1,000 |
| HackerNews | Forum | 1,000 | 1,000 |
| PubMed Central | Medical | 1,000 | 1,000 |
| Wikipedia (en) | Encyclopedia | 1,000 | 1,000 |
| Pile CC | Web Crawl | 1,000 | 1,000 |
| *Standard NLP Tasks* | | | |
| WikiText-103 | Lang. Model | 10,000 | 10,000 |
| AG News | Classification | 10,000 | 10,000 |
| XSum | Summarization | 10,000 | 10,000 |

### D.1.2 Models and Fine-tuning

We investigate two state-of-the-art DLMs: **LLaDA-8B-Base** (Nie et al., 2025) and **Dream-v0-Base-7B** (Ye et al., 2025a). For reference models in calibrated attacks (including SAMA and Ratio), we use the pre-trained versions of these models. Fine-tuning employs AdamW optimizer with learning rate $5 \times 10^{-5}$, weight decay 0.1, batch size 48, and 4 epochs with early stopping (patience 3). Training uses bf16 precision on $3 \times$ NVIDIA A100 GPUs with DeepSpeed Stage 1 optimization (Aminabadi et al., 2022).

### D.1.3 SAMA Configuration

SAMA employs progressive masking with $T = 16$ steps, increasing masking density from $\alpha_{\min} = 5\%$ to $\alpha_{\max} = 50\%$. At each step, we sample $N = 128$ random subsets of $m = 10$ tokens, compute binary indicators based on loss differences, and aggregate using inverse-step weighting. All scores are averaged over 4 Monte Carlo samples for stability.

### D.1.4 MIA Baselines

We compare against twelve baseline MIA methods. For all baselines requiring model queries, scores are averaged over 16 Monte Carlo samples (matching SAMA's progressive steps) to ensure stable estimates under the stochastic nature of diffusion models' masking process.

**Autoregressive-based Baselines Adapted for DLMs.**

**Loss** (Yeom et al., 2018): The most fundamental baseline that directly uses a sample's reconstruction loss as its membership score. For DLMs, we compute the average negative log-likelihood of correctly reconstructing masked tokens across random mask configurations with 15% of tokens masked. Lower reconstruction loss indicates higher membership likelihood, as the model better predicts tokens it has seen during training.

**ZLIB** (Carlini et al., 2021): This method compares the model's reconstruction difficulty against the text's inherent complexity, measured through compression. The membership score is the ratio of the model's loss to the compressed length of the text using ZLIB compression (compression level 6). Lower ratios suggest membership, indicating the model finds the text easier to reconstruct relative to its complexity.

**Lowercase** (Carlini et al., 2021): This baseline exploits the observation that models memorize exact casing patterns during training. The score is computed as the difference between the reconstruction loss on lowercased text and the original text, both evaluated with 15% masking. Positive differences indicate potential membership, suggesting the model is sensitive to the specific casing it memorized.

**Min-K%** (Shi et al., 2024b): Originally designed for autoregressive models, this method focuses on the least confident predictions. For DLMs, we randomly mask 15% of tokens in each of 16 iterations, record the probability assigned to each true token at masked positions, average these probabilities per token across iterations, and sum the smallest 20% of these averaged probabilities. Lower scores indicate membership, as even the model's least confident predictions are relatively accurate for memorized content.

**Min-K%++** (Zhang et al., 2025a): An enhanced version that uses log-probabilities for improved numerical stability. Following the same procedure as Min-K%, we work with log-probabilities of true tokens, summing the 20% most negative values. A small constant is added to probabilities before logarithm computation to prevent numerical errors.

**BoWs** (Das et al., 2025): A query-free method that analyzes textual patterns without accessing the model. Texts are converted to term frequency-inverse document frequency (TF-IDF) representations with 5000 maximum features, ignoring terms appearing in fewer than 5% of documents. A Random Forest classifier with 100 shallow decision trees (maximum depth of 2, minimum 5 samples per leaf) is trained using 5-fold cross-validation. The membership score is the predicted probability of being a training member.

**ReCall** (Xie et al., 2024): This method tests whether prepending non-member context affects reconstruction difficulty. We select 7 non-member text examples from an external dataset, compute the reconstruction loss when these examples precede the target text, and compare to the baseline reconstruction loss without any prefix. The membership score is the ratio of baseline loss to prefixed loss. Higher ratios indicate membership, suggesting the model doesn't benefit from the irrelevant context when reconstructing memorized content.

**CON-ReCall** (Wang et al., 2025): This approach contrasts the effects of member versus non-member prefixes. We compute reconstruction losses using 7 member prefixes (from presumed training data) and 7 non-member prefixes, then calculate the score as the difference between non-member-prefixed and member-prefixed losses, normalized by the baseline loss. Positive scores indicate membership, as the model benefits more from a relevant training context. Prefixes are matched by average length to minimize confounding factors.

**Ratio** (Watson et al., 2021): This calibration method isolates fine-tuning effects by comparing target and reference model losses. The score is the ratio of reference model loss to target model loss. Higher ratios indicate membership, as the fine-tuned model should show lower loss on its training data compared to the reference model.

**Diffusion-specific Baselines.**

**SecMI** (Duan et al., 2023): Originally designed for image diffusion models, SecMI exploits the observation that member samples exhibit lower reconstruction errors across different corruption levels. For DLMs, we adapt this by analyzing token prediction errors across multiple masking densities. The implementation proceeds as follows: We sample 5 masking ratios linearly spaced between 10% and 80% of tokens. At each masking ratio, we apply deterministic masks (seeded by the sample content and step for reproducibility), then compute the cross-entropy loss for predicting the original tokens at masked positions. The errors are aggregated using inverse step weighting, where early steps (lower masking ratios) receive higher weights (weight proportional to 1/(step+1)), as they typically provide cleaner signals. The final score is the negative weighted average of these errors—lower reconstruction error indicates higher membership likelihood, as the model better reconstructs tokens it memorized during training.

**PIA** (Kong et al., 2023): Proximal Initialization Attack analyzes how masking affects the model's prediction consistency. For DLMs, we implement this by comparing model outputs between unmasked and partially masked versions of the same text. The method first obtains the model's predictions on the completely unmasked text (baseline), then creates a masked version with 30% of tokens replaced by mask tokens (using deterministic masking seeded by text hash for reproducibility). We compute the difference between the two sets of predictions at the masked positions using cross-entropy loss difference. The intuition is that for member samples, the model maintains more consistent predictions even when context is partially obscured, resulting in smaller differences. The final score is the negative of this difference—smaller changes in predictions (lower error) indicate

Table 4: Model Utility Comparison on LLaDA-8B-Base: Pre-trained vs. Fine-tuned Stages. Lower Perplexity (PPL, ↓) is better; higher LLM-as-a-judge (LLM Judge, ↑) score is better.

| Dataset | Pre-trained Model | | | Fine-tuned Model | | |
|---|---|---|---|---|---|---|
| | Test PPL (↓) | Train PPL (↓) | LLM Judge (↑) | Test PPL (↓) | Train PPL (↓) | LLM Judge (↑) |
| ArXiv | 13.327 | 13.217 | 0.475 | 12.043 | 9.648 | 0.742 |
| Hacker News | 27.207 | 20.908 | 0.490 | 18.527 | 14.786 | 0.685 |
| Pile (CC) | 17.705 | 17.369 | 0.525 | 17.489 | 14.012 | 0.748 |
| PubMed Central | 12.148 | 12.461 | 0.625 | 10.332 | 8.467 | 0.771 |
| Wikipedia (en) | 8.344 | 7.875 | 0.640 | 8.394 | 6.562 | 0.708 |
| GitHub | 4.320 | 3.641 | 0.506 | 4.378 | 3.241 | 0.593 |
| WikiText-103 | 8.878 | 8.783 | 0.400 | 7.195 | 6.684 | 0.644 |
| AG News | 28.419 | 28.984 | 0.415 | 21.036 | 16.329 | 0.663 |
| XSum | 12.393 | 12.543 | 0.590 | 12.214 | 11.658 | 0.705 |

Table 5: Model Utility Comparison on Dream-v0-Base-7B: Pre-trained vs. Fine-tuned Stages. Lower Perplexity (PPL, ↓) is better; higher LLM-as-a-judge (LLM Judge, ↑) score is better.

| Dataset | Pre-trained Model | | | Fine-tuned Model | | |
|---|---|---|---|---|---|---|
| | Test PPL (↓) | Train PPL (↓) | LLM Judge (↑) | Test PPL (↓) | Train PPL (↓) | LLM Judge (↑) |
| ArXiv | 15.195 | 15.157 | 0.585 | 14.887 | 10.812 | 0.768 |
| Hacker News | 23.769 | 23.186 | 0.455 | 21.924 | 15.036 | 0.671 |
| Pile (CC) | 17.399 | 17.853 | 0.520 | 16.973 | 14.947 | 0.654 |
| PubMed Central | 11.087 | 11.368 | 0.650 | 11.416 | 8.702 | 0.792 |
| Wikipedia (en) | 10.478 | 10.293 | 0.680 | 10.125 | 8.493 | 0.763 |
| GitHub | 6.980 | 6.008 | 0.760 | 5.829 | 5.358 | 0.804 |

higher membership likelihood. Alternative error metrics (L1 or L2 norm between logits) can also be used, though cross-entropy typically provides the most stable signals.

All methods use consistent sequence truncation at 512 tokens and identical random seeds(42) for reproducibility.

### D.1.5 Evaluation Metrics

MIA performance is evaluated using: (1) **AUC**: Area under ROC curve, where 1.0 indicates perfect discrimination; (2) **TPR@FPR**: True positive rates at 10%, 1%, and 0.1% false positive rates, assessing practical privacy risks. Model utility is verified through perplexity on held-out sets and LLM-as-a-Judge evaluation using structured prompts.

## D.2 Performance of Target LLMs

To ensure that our fine-tuned diffusion-based LLMs maintain high utility and are not merely overfitting to the training data, which is a condition that could artificially inflate MIA success (Yeom et al., 2018), we evaluate their performance using two primary approaches: perplexity and an LLM-as-a-Judge framework (Zheng et al., 2023a), as shown in Table 4 and Table 5.

Generally, both Test PPL and Train PPL values decreased after fine-tuning, indicating an enhanced understanding and generation capability of the models. Concurrently, the LLM-as-a-judge scores,

Table 6: Additional MIA performance (AUC, TPR@10%FPR, TPR@1%FPR, TPR@0.1%FPR) across WikiText-103, AG News, and XSum datasets. Each cell shows the mean with std. dev. as a subscript. Best-performing results are highlighted.

| MIAs | WikiText-103 | | | | AG News | | | | XSum | | | |
|---|---|---|---|---|---|---|---|---|---|---|---|---|
| | AUC | T@10% | T@1% | T@0.1% | AUC | T@10% | T@1% | T@0.1% | AUC | T@10% | T@1% | T@0.1% |
| Loss | $0.518_{\pm.01}$ | $0.122_{\pm.02}$ | $0.014_{\pm.01}$ | $0.001_{\pm.00}$ | $0.526_{\pm.01}$ | $0.135_{\pm.01}$ | $0.015_{\pm.00}$ | $0.002_{\pm.00}$ | $0.532_{\pm.01}$ | $0.138_{\pm.01}$ | $0.017_{\pm.01}$ | $0.002_{\pm.00}$ |
| ZLIB | $0.525_{\pm.01}$ | $0.128_{\pm.02}$ | $0.015_{\pm.00}$ | $0.002_{\pm.00}$ | $0.534_{\pm.01}$ | $0.141_{\pm.01}$ | $0.018_{\pm.00}$ | $0.002_{\pm.00}$ | $0.540_{\pm.01}$ | $0.145_{\pm.01}$ | $0.018_{\pm.00}$ | $0.003_{\pm.00}$ |
| Lowercase | $0.536_{\pm.01}$ | $0.136_{\pm.01}$ | $0.017_{\pm.00}$ | $0.002_{\pm.00}$ | $0.542_{\pm.01}$ | $0.148_{\pm.01}$ | $0.019_{\pm.00}$ | $0.003_{\pm.00}$ | $0.548_{\pm.01}$ | $0.152_{\pm.01}$ | $0.020_{\pm.00}$ | $0.003_{\pm.00}$ |
| Min-K% | $0.522_{\pm.01}$ | $0.125_{\pm.01}$ | $0.016_{\pm.00}$ | $0.001_{\pm.00}$ | $0.530_{\pm.01}$ | $0.138_{\pm.01}$ | $0.017_{\pm.00}$ | $0.002_{\pm.00}$ | $0.536_{\pm.01}$ | $0.141_{\pm.01}$ | $0.019_{\pm.01}$ | $0.002_{\pm.00}$ |
| Min-K%++ | $0.519_{\pm.01}$ | $0.123_{\pm.01}$ | $0.014_{\pm.00}$ | $0.001_{\pm.00}$ | $0.527_{\pm.01}$ | $0.136_{\pm.01}$ | $0.016_{\pm.00}$ | $0.002_{\pm.00}$ | $0.533_{\pm.01}$ | $0.139_{\pm.01}$ | $0.017_{\pm.00}$ | $0.002_{\pm.00}$ |
| BoWs | $0.501_{\pm.01}$ | $0.105_{\pm.01}$ | $0.010_{\pm.00}$ | $0.000_{\pm.00}$ | $0.508_{\pm.01}$ | $0.118_{\pm.01}$ | $0.012_{\pm.00}$ | $0.001_{\pm.00}$ | $0.515_{\pm.01}$ | $0.122_{\pm.01}$ | $0.013_{\pm.00}$ | $0.001_{\pm.00}$ |
| ReCall | $0.521_{\pm.01}$ | $0.124_{\pm.02}$ | $0.014_{\pm.00}$ | $0.001_{\pm.00}$ | $0.529_{\pm.01}$ | $0.137_{\pm.01}$ | $0.016_{\pm.00}$ | $0.002_{\pm.00}$ | $0.535_{\pm.01}$ | $0.140_{\pm.01}$ | $0.017_{\pm.00}$ | $0.002_{\pm.00}$ |
| CON-Recall | $0.517_{\pm.01}$ | $0.121_{\pm.01}$ | $0.013_{\pm.00}$ | $0.001_{\pm.00}$ | $0.525_{\pm.01}$ | $0.134_{\pm.01}$ | $0.015_{\pm.00}$ | $0.002_{\pm.00}$ | $0.531_{\pm.01}$ | $0.137_{\pm.01}$ | $0.016_{\pm.00}$ | $0.002_{\pm.00}$ |
| Neighbor | $0.508_{\pm.01}$ | $0.113_{\pm.01}$ | $0.011_{\pm.00}$ | $0.001_{\pm.00}$ | $0.516_{\pm.01}$ | $0.126_{\pm.01}$ | $0.014_{\pm.00}$ | $0.001_{\pm.00}$ | $0.522_{\pm.01}$ | $0.129_{\pm.01}$ | $0.015_{\pm.00}$ | $0.001_{\pm.00}$ |
| Ratio | $0.584_{\pm.01}$ | $0.178_{\pm.02}$ | $0.028_{\pm.01}$ | $0.003_{\pm.00}$ | $0.608_{\pm.01}$ | $0.205_{\pm.02}$ | $0.034_{\pm.01}$ | $0.004_{\pm.00}$ | $0.616_{\pm.01}$ | $0.212_{\pm.02}$ | $0.036_{\pm.01}$ | $0.005_{\pm.00}$ |
| SecMI | $0.528_{\pm.01}$ | $0.130_{\pm.01}$ | $0.015_{\pm.00}$ | $0.001_{\pm.00}$ | $0.536_{\pm.01}$ | $0.143_{\pm.01}$ | $0.017_{\pm.00}$ | $0.002_{\pm.00}$ | $0.542_{\pm.01}$ | $0.147_{\pm.01}$ | $0.018_{\pm.00}$ | $0.002_{\pm.00}$ |
| PIA | $0.524_{\pm.01}$ | $0.127_{\pm.01}$ | $0.014_{\pm.00}$ | $0.001_{\pm.00}$ | $0.532_{\pm.01}$ | $0.140_{\pm.01}$ | $0.016_{\pm.00}$ | $0.002_{\pm.00}$ | $0.538_{\pm.01}$ | $0.144_{\pm.01}$ | $0.017_{\pm.00}$ | $0.002_{\pm.00}$ |
| SAMA (Ours) | $\mathbf{0.782_{\pm.01}}$ | $\mathbf{0.415_{\pm.03}}$ | $\mathbf{0.108_{\pm.02}}$ | $\mathbf{0.018_{\pm.01}}$ | $\mathbf{0.673_{\pm.01}}$ | $\mathbf{0.268_{\pm.02}}$ | $\mathbf{0.052_{\pm.01}}$ | $\mathbf{0.007_{\pm.00}}$ | $\mathbf{0.682_{\pm.01}}$ | $\mathbf{0.277_{\pm.02}}$ | $\mathbf{0.055_{\pm.01}}$ | $\mathbf{0.008_{\pm.00}}$ |

which reflect the quality of the model's output as assessed by another LLM, generally increased post-fine-tuning. This trend was observed across diverse datasets, including those from the MIMIR benchmark and standard NLP benchmarks such as ArXiv, DM Mathematics, Hacker News, Pile (CC), PubMed Central, Wikipedia (en), GitHub, WikiText-103, AG News, and XSum. These observations collectively suggest that the fine-tuning stage effectively refined the models' abilities.

## D.3 Extended Main Comparison Results

This section provides a more extensive evaluation of SAMA, detailing its performance with the LLaDA-8B-Base model on additional benchmark datasets including WikiText-103, AG News, and XSum. It also presents comprehensive findings for the Dream-v0-Base-7B model across six diverse datasets. These supplementary results offer a broader perspective on SAMA's robust performance and the privacy implications for diffusion-based LLMs.

For the LLaDA-8B-Base model, further analysis is presented in Table 6. On WikiText-103, SAMA achieves an AUC of 0.782 with a TPR@1%FPR of 0.108, substantially outperforming the second-best method (Ratio), which achieves 0.584 AUC. For AG News and XSum datasets, SAMA maintains strong performance with AUCs of 0.673 and 0.682, respectively, though with more modest TPR@1%FPR values of 0.052 and 0.055. While the improvement margin is less dramatic than on other datasets, SAMA still consistently outperforms all baselines by approximately 10-20% in AUC, demonstrating its effectiveness across diverse text domains.

Turning to the Dream-v0-Base-7B model, Table 7 reveals strong MIA performance by SAMA despite the model's diffusion-based architecture. SAMA achieves exceptional performance on GitHub with an AUC of 0.806 and TPR@1%FPR of 0.168, representing a 45% improvement over the second-best method (ZLIB at 0.558 AUC). On Wikipedia (en), SAMA attains an AUC of 0.764 with TPR@1%FPR of 0.112, demonstrating robust attack efficacy. For ArXiv and Pile CC, SAMA achieves AUCs of 0.748 and 0.745, respectively, with corresponding TPR@1%FPR values of 0.098 and 0.096. PubMed Central shows an AUC of 0.732 with TPR@1%FPR of 0.081, while HackerNews exhibits an AUC of 0.615 with TPR@1%FPR of 0.024.

## D.4 Extended Ablation Studies

To assess the robustness of SAMA and the contribution of its hyperparameters, we conduct a sensitivity analysis on the LLaDA-8B-Base model using the ArXiv dataset. We vary three key parameters: the number of progressive masking steps ($T$), the subset size ($m$), and the number of sampled subsets ($N$). Figure 4 illustrates the impact of these variations on AUC and TPR@1%FPR.

Table 7: MIA performance on Dream-v0-Base-7B model across different datasets. Each cell shows the mean with std. dev. as a subscript. Best-performing results are highlighted. Note the more irregular performance patterns compared to LLaDA models.

| MIAs | ArXiv | | | | GitHub | | | | HackerNews | | | |
|---|---|---|---|---|---|---|---|---|---|---|---|---|
| | AUC | T@10% | T@1% | T@0.1% | AUC | T@10% | T@1% | T@0.1% | AUC | T@10% | T@1% | T@0.1% |
| Loss | $0.498_{\pm.02}$ | $0.103_{\pm.03}$ | $0.009_{\pm.01}$ | $0.000_{\pm.00}$ | $0.524_{\pm.01}$ | $0.128_{\pm.02}$ | $0.018_{\pm.01}$ | $0.002_{\pm.00}$ | $0.491_{\pm.02}$ | $0.095_{\pm.02}$ | $0.008_{\pm.01}$ | $0.000_{\pm.00}$ |
| ZLIB | $0.485_{\pm.02}$ | $0.091_{\pm.02}$ | $0.007_{\pm.00}$ | $0.000_{\pm.00}$ | $0.558_{\pm.02}$ | $0.168_{\pm.03}$ | $0.028_{\pm.01}$ | $0.004_{\pm.00}$ | $0.507_{\pm.01}$ | $0.112_{\pm.02}$ | $0.012_{\pm.01}$ | $0.001_{\pm.00}$ |
| Lowercase | $0.509_{\pm.01}$ | $0.097_{\pm.02}$ | $0.010_{\pm.01}$ | $0.001_{\pm.00}$ | $0.498_{\pm.02}$ | $0.104_{\pm.02}$ | $0.014_{\pm.01}$ | $0.001_{\pm.00}$ | $0.482_{\pm.02}$ | $0.086_{\pm.01}$ | $0.007_{\pm.00}$ | $0.001_{\pm.00}$ |
| Min-K% | $0.532_{\pm.02}$ | $0.125_{\pm.02}$ | $0.015_{\pm.01}$ | $0.001_{\pm.00}$ | $0.515_{\pm.01}$ | $0.119_{\pm.02}$ | $0.021_{\pm.01}$ | $0.003_{\pm.00}$ | $0.518_{\pm.01}$ | $0.121_{\pm.02}$ | $0.016_{\pm.01}$ | $0.002_{\pm.00}$ |
| Min-K%++ | $0.504_{\pm.02}$ | $0.106_{\pm.02}$ | $0.009_{\pm.00}$ | $0.000_{\pm.00}$ | $0.483_{\pm.02}$ | $0.094_{\pm.01}$ | $0.011_{\pm.01}$ | $0.001_{\pm.00}$ | $0.499_{\pm.01}$ | $0.108_{\pm.02}$ | $0.011_{\pm.00}$ | $0.001_{\pm.00}$ |
| BoWs | $0.519_{\pm.01}$ | $0.107_{\pm.01}$ | $0.011_{\pm.00}$ | $0.000_{\pm.00}$ | $0.656_{\pm.02}$ | $0.306_{\pm.02}$ | $0.154_{\pm.02}$ | $0.059_{\pm.05}$ | $0.527_{\pm.01}$ | $0.128_{\pm.01}$ | $0.009_{\pm.00}$ | $0.001_{\pm.00}$ |
| ReCall | $0.501_{\pm.01}$ | $0.109_{\pm.02}$ | $0.009_{\pm.01}$ | $0.000_{\pm.00}$ | $0.531_{\pm.02}$ | $0.142_{\pm.02}$ | $0.023_{\pm.01}$ | $0.003_{\pm.00}$ | $0.496_{\pm.01}$ | $0.102_{\pm.01}$ | $0.009_{\pm.01}$ | $0.000_{\pm.00}$ |
| CON-Recall | $0.494_{\pm.02}$ | $0.098_{\pm.02}$ | $0.008_{\pm.00}$ | $0.000_{\pm.00}$ | $0.523_{\pm.02}$ | $0.134_{\pm.02}$ | $0.019_{\pm.01}$ | $0.002_{\pm.00}$ | $0.488_{\pm.02}$ | $0.092_{\pm.01}$ | $0.008_{\pm.01}$ | $0.000_{\pm.00}$ |
| Neighbor | $0.506_{\pm.02}$ | $0.101_{\pm.01}$ | $0.011_{\pm.01}$ | $0.001_{\pm.00}$ | $0.479_{\pm.02}$ | $0.082_{\pm.02}$ | $0.007_{\pm.01}$ | $0.000_{\pm.00}$ | $0.546_{\pm.02}$ | $0.148_{\pm.02}$ | $0.021_{\pm.01}$ | $0.001_{\pm.00}$ |
| Ratio | $0.521_{\pm.02}$ | $0.118_{\pm.02}$ | $0.016_{\pm.01}$ | $0.001_{\pm.00}$ | $0.549_{\pm.02}$ | $0.162_{\pm.03}$ | $0.029_{\pm.01}$ | $0.004_{\pm.00}$ | $0.535_{\pm.02}$ | $0.141_{\pm.02}$ | $0.018_{\pm.01}$ | $0.001_{\pm.00}$ |
| SecMI | $0.513_{\pm.02}$ | $0.108_{\pm.03}$ | $0.010_{\pm.01}$ | $0.000_{\pm.00}$ | $0.537_{\pm.02}$ | $0.149_{\pm.02}$ | $0.025_{\pm.01}$ | $0.004_{\pm.01}$ | $0.511_{\pm.03}$ | $0.117_{\pm.02}$ | $0.013_{\pm.00}$ | $0.001_{\pm.00}$ |
| PIA | $0.502_{\pm.01}$ | $0.096_{\pm.01}$ | $0.008_{\pm.01}$ | $0.000_{\pm.00}$ | $0.509_{\pm.01}$ | $0.113_{\pm.01}$ | $0.013_{\pm.00}$ | $0.001_{\pm.00}$ | $0.486_{\pm.02}$ | $0.090_{\pm.01}$ | $0.009_{\pm.00}$ | $0.000_{\pm.00}$ |
| SAMA (Ours) | $\mathbf{0.748_{\pm.01}}$ | $\mathbf{0.412_{\pm.03}}$ | $\mathbf{0.098_{\pm.02}}$ | $\mathbf{0.011_{\pm.01}}$ | $\mathbf{0.806_{\pm.01}}$ | $\mathbf{0.498_{\pm.03}}$ | $\mathbf{0.168_{\pm.03}}$ | $\mathbf{0.042_{\pm.02}}$ | $\mathbf{0.615_{\pm.01}}$ | $\mathbf{0.186_{\pm.02}}$ | $\mathbf{0.024_{\pm.01}}$ | $\mathbf{0.003_{\pm.00}}$ |

| MIAs | PubMed Central | | | | Wikipedia (en) | | | | Pile CC | | | |
|---|---|---|---|---|---|---|---|---|---|---|---|---|
| | AUC | T@10% | T@1% | T@0.1% | AUC | T@10% | T@1% | T@0.1% | AUC | T@10% | T@1% | T@0.1% |
| Loss | $0.489_{\pm.01}$ | $0.096_{\pm.02}$ | $0.010_{\pm.01}$ | $0.001_{\pm.00}$ | $0.493_{\pm.02}$ | $0.089_{\pm.01}$ | $0.009_{\pm.00}$ | $0.001_{\pm.00}$ | $0.497_{\pm.01}$ | $0.101_{\pm.02}$ | $0.011_{\pm.01}$ | $0.001_{\pm.00}$ |
| ZLIB | $0.481_{\pm.02}$ | $0.088_{\pm.01}$ | $0.006_{\pm.00}$ | $0.000_{\pm.00}$ | $0.519_{\pm.01}$ | $0.124_{\pm.02}$ | $0.017_{\pm.01}$ | $0.002_{\pm.00}$ | $0.488_{\pm.01}$ | $0.093_{\pm.01}$ | $0.008_{\pm.00}$ | $0.000_{\pm.00}$ |
| Lowercase | $0.502_{\pm.01}$ | $0.093_{\pm.01}$ | $0.004_{\pm.00}$ | $0.000_{\pm.00}$ | $0.509_{\pm.02}$ | $0.099_{\pm.02}$ | $0.010_{\pm.00}$ | $0.001_{\pm.00}$ | $0.514_{\pm.01}$ | $0.103_{\pm.01}$ | $0.009_{\pm.00}$ | $0.001_{\pm.00}$ |
| Min-K% | $0.496_{\pm.01}$ | $0.109_{\pm.02}$ | $0.009_{\pm.00}$ | $0.001_{\pm.00}$ | $0.483_{\pm.01}$ | $0.077_{\pm.01}$ | $0.008_{\pm.00}$ | $0.000_{\pm.00}$ | $0.529_{\pm.02}$ | $0.131_{\pm.02}$ | $0.017_{\pm.01}$ | $0.002_{\pm.00}$ |
| Min-K%++ | $0.491_{\pm.01}$ | $0.103_{\pm.01}$ | $0.008_{\pm.00}$ | $0.000_{\pm.00}$ | $0.501_{\pm.01}$ | $0.107_{\pm.01}$ | $0.007_{\pm.00}$ | $0.000_{\pm.00}$ | $0.490_{\pm.01}$ | $0.098_{\pm.01}$ | $0.007_{\pm.00}$ | $0.001_{\pm.00}$ |
| BoWs | $0.489_{\pm.01}$ | $0.100_{\pm.01}$ | $0.002_{\pm.00}$ | $0.000_{\pm.00}$ | $0.471_{\pm.01}$ | $0.084_{\pm.02}$ | $0.003_{\pm.00}$ | $0.001_{\pm.00}$ | $0.480_{\pm.01}$ | $0.092_{\pm.01}$ | $0.003_{\pm.00}$ | $0.001_{\pm.00}$ |
| ReCall | $0.493_{\pm.01}$ | $0.090_{\pm.01}$ | $0.007_{\pm.00}$ | $0.000_{\pm.00}$ | $0.504_{\pm.01}$ | $0.101_{\pm.01}$ | $0.009_{\pm.01}$ | $0.000_{\pm.00}$ | $0.498_{\pm.01}$ | $0.095_{\pm.01}$ | $0.008_{\pm.00}$ | $0.000_{\pm.00}$ |
| CON-Recall | $0.518_{\pm.02}$ | $0.117_{\pm.02}$ | $0.012_{\pm.01}$ | $0.001_{\pm.00}$ | $0.493_{\pm.02}$ | $0.092_{\pm.01}$ | $0.008_{\pm.00}$ | $0.000_{\pm.00}$ | $0.496_{\pm.01}$ | $0.100_{\pm.01}$ | $0.009_{\pm.00}$ | $0.000_{\pm.00}$ |
| Neighbor | $0.507_{\pm.01}$ | $0.097_{\pm.01}$ | $0.010_{\pm.01}$ | $0.000_{\pm.00}$ | $0.527_{\pm.02}$ | $0.118_{\pm.02}$ | $0.018_{\pm.01}$ | $0.002_{\pm.00}$ | $0.502_{\pm.01}$ | $0.093_{\pm.01}$ | $0.009_{\pm.00}$ | $0.000_{\pm.00}$ |
| Ratio | $0.514_{\pm.02}$ | $0.114_{\pm.02}$ | $0.011_{\pm.01}$ | $0.001_{\pm.00}$ | $0.523_{\pm.02}$ | $0.121_{\pm.02}$ | $0.014_{\pm.01}$ | $0.001_{\pm.00}$ | $0.526_{\pm.01}$ | $0.128_{\pm.01}$ | $0.015_{\pm.01}$ | $0.001_{\pm.00}$ |
| SecMI | $0.501_{\pm.02}$ | $0.102_{\pm.02}$ | $0.009_{\pm.00}$ | $0.000_{\pm.00}$ | $0.514_{\pm.02}$ | $0.109_{\pm.01}$ | $0.011_{\pm.01}$ | $0.001_{\pm.00}$ | $0.508_{\pm.02}$ | $0.106_{\pm.02}$ | $0.010_{\pm.00}$ | $0.001_{\pm.00}$ |
| PIA | $0.488_{\pm.01}$ | $0.092_{\pm.01}$ | $0.006_{\pm.00}$ | $0.000_{\pm.00}$ | $0.507_{\pm.01}$ | $0.103_{\pm.01}$ | $0.009_{\pm.00}$ | $0.000_{\pm.00}$ | $0.501_{\pm.01}$ | $0.097_{\pm.01}$ | $0.008_{\pm.00}$ | $0.000_{\pm.00}$ |
| SAMA (Ours) | $\mathbf{0.732_{\pm.01}}$ | $\mathbf{0.338_{\pm.02}}$ | $\mathbf{0.081_{\pm.01}}$ | $\mathbf{0.009_{\pm.00}}$ | $\mathbf{0.764_{\pm.01}}$ | $\mathbf{0.426_{\pm.03}}$ | $\mathbf{0.112_{\pm.02}}$ | $\mathbf{0.016_{\pm.01}}$ | $\mathbf{0.745_{\pm.01}}$ | $\mathbf{0.382_{\pm.02}}$ | $\mathbf{0.096_{\pm.01}}$ | $\mathbf{0.011_{\pm.00}}$ |

**Impact of Progressive Steps** ($T$). We observe a strong positive correlation between the number of masking steps and attack performance. Increasing $T$ from 1 to 16 yields a significant improvement, raising the AUC from 0.741 to 0.805 and nearly doubling the TPR@1%FPR from 0.089 to 0.174. While extending $T$ further to 48 achieves even higher performance (AUC 0.845), we select $T = 16$ as the default configuration to balance attack efficacy with query complexity.

**Impact of Subset Size** ($m$). The size of the token subset plays a critical role in robust aggregation. Extremely small subsets (e.g., $m = 1$) yield suboptimal performance (AUC 0.783), as they lack sufficient context to distinguish membership signals from noise. Performance improves as $m$ increases, reaching an AUC of 0.856 at $m = 32$. This suggests that larger subsets can effectively capture coherent membership patterns in long-context domains like ArXiv. However, our default choice of $m = 10$ remains a robust operating point that significantly outperforms the baseline.

**Impact of Number of Subsets** ($N$). SAMA demonstrates a saturation effect with respect to the number of sampled subsets $N$. Performance improves rapidly as $N$ increases from 1 (AUC 0.755) to 32 (AUC 0.803), demonstrating the necessity of sufficient sampling to estimate the robust mean. However, beyond $N = 32$, the performance plateaus, with $N = 128$ yielding a marginal gain to 0.805. This indicates that the sign-based statistic converges quickly. Since these subsets are sampled offline with negligible computational cost, we employ $N = 128$ to ensure maximum robustness without incurring additional model queries.

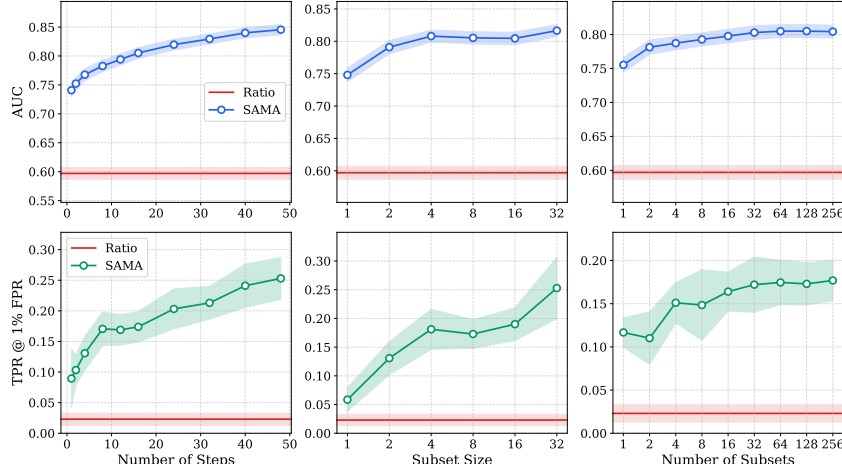

Figure 4: Sensitivity analysis of SAMA hyperparameters on LLaDA-8B-Base (ArXiv). The top row reports AUC, and the bottom row reports TPR@1%FPR. We vary the number of progressive steps $T$ (left), subset size $m$ (middle), and number of subsets $N$ (right). The dashed red line indicates the performance of the best baseline (Ratio).

Table 8: Comparison of LoRA configurations and DP-LoRA privacy levels on ArXiv. **Left:** LoRA rank ($r$) impact. **Right:** DP-LoRA with varying privacy budget ($\epsilon$). Lower perplexity indicates better utility; higher AUC indicates stronger membership inference.

| $r$ | Trn. % | PPL | AUC | T@10% |
|---|---|---|---|---|
| 256 | 4.02% | 13.27 | 0.528 | 0.106 |
| 512 | 7.73% | 12.95 | 0.571 | 0.142 |
| 1024 | 14.34% | 12.61 | 0.638 | 0.218 |
| 2048 | 25.09% | 12.38 | 0.726 | 0.321 |
| Full | 100.00% | **12.04** | **0.850** | **0.586** |

| $\epsilon$ | PPL | AUC | T@10% | T@1% |
|---|---|---|---|---|
| 0.01 | 13.46 | 0.497 | 0.098 | 0.010 |
| 1.0 | 13.18 | 0.502 | 0.109 | 0.012 |
| 10.0 | 12.74 | 0.548 | 0.128 | 0.015 |
| 20.0 | 12.35 | 0.607 | 0.176 | 0.024 |
| N/A | **12.04** | **0.850** | **0.586** | **0.178** |

## D.5 Defending against SAMA

We evaluate two parameter-efficient fine-tuning methods as potential defenses against SAMA: standard Low-Rank Adaptation (LoRA) and its differentially private variant (DP-LoRA). These methods naturally limit memorization by constraining the parameter space during fine-tuning, potentially reducing vulnerability to membership inference. We analyze the privacy-utility trade-offs of both approaches on the ArXiv dataset, demonstrating that while both defenses can significantly reduce SAMA's effectiveness, they require careful calibration to maintain acceptable model utility.

### D.5.1 Low-Rank Adaptation (LoRA)

To evaluate defense strategies against SAMA, we investigate Low-Rank Adaptation (LoRA) Hu et al. (2021) fine-tuning on the LLaDA-8B-Base model using the ArXiv dataset. LoRA reduces the number of trainable parameters through low-rank decomposition (with rank $r$ and $\alpha = 2r$), potentially limiting memorization—a key vulnerability exploited by membership inference attacks (Luo et al., 2024; Amit et al., 2024; Zhang et al., 2025b).

Table 8 (left) reveals a clear privacy-utility trade-off. Lower ranks provide stronger privacy protection: at $r = 256$ (4.02% trainable parameters), SAMA's AUC drops substantially from 0.850 (full fine-tuning) to 0.528, with TPR@10%FPR decreasing from 0.586 to 0.106. This privacy gain, however, incurs a utility cost—perplexity increases from 12.04 to 13.27. As the rank increases, model utility improves, but privacy protection weakens. At $r = 2048$ (25.09% trainable parameters), per-

Table 9: Impact of temperature scaling on membership inference attacks. Higher temperatures increase output entropy, providing modest defense with no retraining required.

| $T$ | Method | ArXiv | | | |
|-----|--------|-------|--------|--------|---------|
| | | AUC | TPR@10% | TPR@1% | TPR@0.1% |
| 1.0 | Ratio | $0.597_{\pm.01}$ | $0.181_{\pm.01}$ | $0.023_{\pm.01}$ | $0.001_{\pm.00}$ |
| | SAMA | $\mathbf{0.850}_{\pm.01}$ | $\mathbf{0.586}_{\pm.03}$ | $\mathbf{0.178}_{\pm.03}$ | $\mathbf{0.014}_{\pm.01}$ |
| 1.1 | Ratio | $0.588_{\pm.01}$ | $0.172_{\pm.01}$ | $0.021_{\pm.01}$ | $0.001_{\pm.00}$ |
| | SAMA | $\mathbf{0.842}_{\pm.01}$ | $\mathbf{0.575}_{\pm.02}$ | $\mathbf{0.175}_{\pm.02}$ | $\mathbf{0.014}_{\pm.01}$ |
| 1.2 | Ratio | $0.575_{\pm.01}$ | $0.163_{\pm.02}$ | $0.019_{\pm.01}$ | $0.001_{\pm.00}$ |
| | SAMA | $\mathbf{0.831}_{\pm.01}$ | $\mathbf{0.560}_{\pm.02}$ | $\mathbf{0.169}_{\pm.02}$ | $\mathbf{0.012}_{\pm.01}$ |
| 1.5 | Ratio | $0.534_{\pm.01}$ | $0.135_{\pm.02}$ | $0.014_{\pm.01}$ | $0.000_{\pm.00}$ |
| | SAMA | $\mathbf{0.776}_{\pm.01}$ | $\mathbf{0.448}_{\pm.03}$ | $\mathbf{0.108}_{\pm.02}$ | $\mathbf{0.005}_{\pm.01}$ |

plexity improves to 12.38, yet SAMA's effectiveness rises significantly (AUC 0.726, TPR@10%FPR 0.321). The intermediate configuration ($r = 1024$) offers a balanced compromise with AUC 0.638 and perplexity 12.61, suggesting that practitioners must carefully calibrate LoRA rank based on their specific privacy-utility requirements.

### D.5.2 Differentially Private LoRA (DP-LoRA)

Differentially Private LoRA (DP-LoRA) Liu et al. (2024) provides formal privacy guarantees against MIAs like SAMA. Computing per-sample gradient norms for gradient clipping—essential for differential privacy—becomes computationally prohibitive for billion-parameter models. DP-LoRA addresses this by restricting per-sample gradient operations, noise injection, and privacy accounting to the compact LoRA adapter parameters. We employ LoRA rank $r = 1024$ (with $\alpha = 2048$, dropout 0.05), apply DP with target $\delta = 1 \times 10^{-5}$, gradient norm clipping at 1.0, and vary the privacy budget $\epsilon$.

Table 8 (right) demonstrates DP-LoRA's effectiveness against SAMA on ArXiv. The baseline (N/A row) represents standard LoRA without differential privacy. Stringent privacy guarantees ($\epsilon = 0.01$) virtually neutralize SAMA, reducing AUC from 0.850 to 0.497—approaching random chance—with TPR@10%FPR plummeting from 0.586 to 0.098. This robust defense incurs a utility penalty, increasing perplexity to 13.46. As $\epsilon$ relaxes, the privacy-utility trade-off becomes more favorable: at $\epsilon = 1.0$, strong privacy persists (AUC 0.512) with improved utility (PPL 13.18), while $\epsilon = 10.0$ achieves AUC 0.548 and PPL 12.74. Even with relaxed privacy ($\epsilon = 20.0$), SAMA's effectiveness (AUC 0.607, TPR@10%FPR 0.176) remains substantially below the unprotected baseline, while perplexity (12.35) approaches baseline performance.

These results confirm that DP-LoRA offers scalable, practical defense for diffusion-based LLMs, effectively mitigating membership inference risks across various privacy budgets. The persistent protective effect even at higher $\epsilon$ values demonstrates DP-LoRA's robustness, though optimal deployment requires careful calibration of the privacy budget to balance protection against SAMA with acceptable model performance for the target application.

### D.5.3 Temperature Scaling

Temperature scaling is often proposed as a low-cost, inference-time defense that flattens the output distribution via $p_i = \exp(z_i/T) / \sum_j \exp(z_j/T)$ (Guo et al., 2017). However, our evaluation in Table 9 reveals that this method fails to offer a practical trade-off between security and utility.

At moderate levels ($T \in [1.1, 1.2]$), the defense offers a false sense of security. The attack performance remains stubbornly high, with SAMA's AUC dropping only marginally from 0.850 to 0.831 (a decrease of merely 2.2%). This resilience stems from the nature of our window-based aggrega-

Table 10: Attack performance against SOFT defense on ArXiv dataset. SOFT configured with selection ratio $\alpha = 0.3$ and paraphrase ratio $\beta = 0.5$.

| Defense | PPL | Method | ArXiv | | | |
|---|---|---|---|---|---|---|
| | | | AUC | TPR@10% | TPR@1% | TPR@0.1% |
| Baseline | 9.648 | Ratio | $0.597_{\pm.01}$ | $0.181_{\pm.01}$ | $0.023_{\pm.01}$ | $0.001_{\pm.00}$ |
| | | **SAMA** | $\mathbf{0.850}_{\pm.01}$ | $\mathbf{0.586}_{\pm.03}$ | $\mathbf{0.178}_{\pm.03}$ | $\mathbf{0.014}_{\pm.01}$ |
| SOFT | 9.647 | Ratio | $0.497_{\pm.003}$ | $0.102_{\pm.004}$ | $0.011_{\pm.001}$ | $0.001_{\pm.000}$ |
| | | **SAMA** | $\mathbf{0.499}_{\pm.004}$ | $\mathbf{0.104}_{\pm.003}$ | $\mathbf{0.013}_{\pm.002}$ | $\mathbf{0.002}_{\pm.000}$ |

tion: because the entropy increase is applied uniformly, the relative ordering of loss values within local windows persists, preserving the differential signal between target and reference models.

When scaling is pushed aggressively to $T = 1.5$, we observe a more tangible reduction in attack success. AUC falls by 8.7% to 0.776, and TPR@1%FPR decreases by approximately 39% (0.178 to 0.108). However, this regime is known to severely degrade generation quality, causing incoherence and repetition. Furthermore, SAMA proves far more robust to this perturbation than the global baseline. As $T$ increases to 1.5, the relative AUC advantage of our method over Ratio actually widens (from $1.42\times$ to $1.45\times$), confirming that local aggregation is structurally better equipped to survive entropy-based obfuscation than global averaging.

### D.5.4 Selective Data Obfuscation in LLM Fine-Tuning

We further evaluate the resilience of SAMA against the SOFT defense (Zhang et al., 2025b), which mitigates privacy risks by identifying and paraphrasing the most "memorable" training samples. Table 10 illustrates a complete collapse in attack efficacy on the ArXiv dataset. While SAMA initially achieves a dominant AUC of 0.850, the application of SOFT suppresses this score to 0.499, effectively rendering the attack no better than a random guess. This demonstrates that by altering the syntax of influential records (approximately 15% of the data) without changing their semantics, SOFT successfully eradicates the specific token-level artifacts that SAMA exploits, all while maintaining the model's general utility.

### D.6 Impact of Reference Model Misalignment

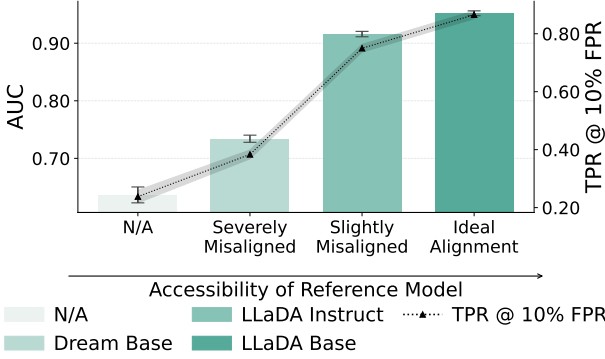

Figure 5: Impact of Reference Model Misalignment on SAMA Performance. Performance (AUC and TPR@10%FPR) is shown for an ideal reference (LLaDA-8B-Base), a slightly misaligned reference (LLaDA-8B-Instruct), and a severely misaligned reference (Dream-v0-7B-Base). Increasing misalignment leads to a reduction in attack performance, though SAMA still outperforms baselines even with severe misalignment.

This section investigates the robustness of SAMA when this ideal reference is unavailable, necessitating the use of alternative, potentially misaligned reference models. We maintain the LLaDA-8B-Base model fine-tuned on the ArXiv dataset as the target and evaluate SAMA's performance using three distinct reference models: the ideal LLaDA-8B-Base (Ideal Alignment), the closely related LLaDA-8B-Instruct (Slightly Misaligned), and the substantially different Dream-v0-7B-Base (Severely Misaligned). LLaDA-8B-Instruct shares architecture and tokenizer with the target but differs in instruction-tuning, affecting utility and prior knowledge. Dream-v0-7B-Base differs more significantly in architecture, pre-training data, and tokenizer.

The results, presented in Figure 5, reveal that SAMA's performance degrades gracefully with increasing reference model misalignment. The slightly misaligned LLaDA-8B-Instruct reference causes a small drop in attack efficacy compared to the ideal LLaDA-8B-Base reference. However, the severely misaligned Dream-v0-7B-Base reference leads to a more critical reduction in performance, although the attack remains more effective than uncalibrated baselines. This performance drop can be attributed to two main factors. Firstly, greater divergence in the reference model's prior knowledge compared to the target model diminishes the accuracy of difficulty calibration. Secondly, differences in tokenization between reference and target models mean that masks applied during each step may cover different semantic units of the sentence, leading to reconstruction losses that reflect disparate aspects of the text and thus less effective calibration.

## D.7 Computational Overhead

We provide a theoretical analysis of the computational cost of SAMA based on the algorithmic specification in Section 3.5, followed by an empirical validation.

**Theoretical Analysis.** The computational complexity of SAMA is dominated by the model inference steps required to extract membership evidence. As detailed in Algorithm 1 (Phase I), the algorithm iterates through $T$ progressive masking steps. Within this outer loop (lines 4-15), the target and reference models are queried exactly once per step (lines 9-10) to compute the loss vectors $\ell^{\mathrm{T}}$ and $\ell^{\mathrm{R}}$ for the masked sequence $\mathbf{x}_{\mathrm{m}}$. The subsequent robust subset aggregation involves an inner loop (lines 11-14) that samples $N$ subsets and computes arithmetic differences. This inner loop operates entirely on the cached loss vectors derived from the single pair of forward passes. Consequently, the operations involving $N$, as well as the sign-based aggregation and adaptive weighting in Phase II (Algorithm 2), are purely CPU-bound arithmetic operations with negligible cost compared to the GPU-bound model inference. Therefore, the total query complexity of SAMA is $\mathcal{O}(T \cdot \mathcal{C}_{fwd})$, where $\mathcal{C}_{fwd}$ is the cost of a forward pass. This asymptotic complexity is identical to standard query-based baselines (such as Loss or Ratio), which also require aggregating results over $T$ Monte Carlo samples to estimate the expected loss over the stochastic masking space of a DLM.

**Empirical Runtime Comparison.** To validate this theoretical assessment, we measured the wall-clock time for performing attacks on the LLaDA-8B-Base model using the ArXiv dataset. To ensure a direct comparison, we fixed the query budget to $T = 16$ for both SAMA and the query-based baselines (Loss and Ratio). As shown in Table 11, the additional CPU overhead from our subset aggregation logic is practically undetectable. The runtime of SAMA (1h 32m 16s) is nearly identical to that of the Ratio baseline (1h 32m 00s). In contrast, methods that require generating new text samples, such as Neighborhood-based attacks, incur significantly higher computational costs, taking over 8 hours to complete the same task. Furthermore, as shown in Section D.4, SAMA remains effective even at lower query budgets. With $T = 4$, SAMA achieves a TPR@1%FPR of 0.1308 while reducing the runtime to approximately 23 minutes on the ArXiv dataset, effectively outperforming the full-budget ($T = 16$) baselines while being 2× faster.

## E The Structure of Membership Signals: An Empirical Analysis

In this section, we provide a plausible explanation for the design choice of SAMA. We conducted an analysis of token-level loss differences. We utilized the LLaDA-8B model fine-tuned on the ArXiv

Table 11: Runtime comparison of MIA methods on LLaDA-8B-Base (fixed query budget $T = 16$ for query-based methods). SAMA incurs negligible overhead compared to the Ratio baseline.

| Dataset | Loss | Ratio | SAMA (Ours) | Neighborhood |
|---|---|---|---|---|
| Wikipedia (en) | 39m 01s | 1h 32m 00s | 1h 32m 16s | 8h 43m 27s |
| PubMed Central | 55m 17s | 1h 37m 04s | 1h 36m 19s | 9h 32m 38s |
| HackerNews | 54m 20s | 1h 35m 08s | 1h 35m 36s | 9h 13m 02s |

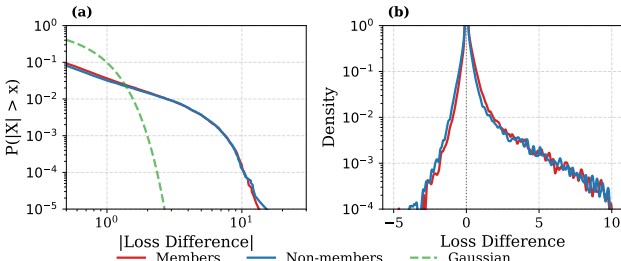

Figure 6: Empirical distribution of token-level loss differences. (a) Complementary CDF confirms the long-tailed behavior of the token loss. (b) Log-scale density plots reveal long-tailed distributions with a subtle rightward shift for members. The difference is most pronounced in the consistent shift of the bulk, whereas the extreme right tail is dominated by shared domain adaptation effects.

subset of the MIMIR benchmark, evaluating both the fine-tuned target model $\mathcal{M}^T_{DF}$ and the pre-trained reference model $\mathcal{M}^R_{DF}$ on balanced sets of 1,000 member samples and 1,000 non-member samples. For each masked sequence, we computed the token-level loss difference $\Delta = \ell^R - \ell^T$, where positive values indicate that the fine-tuned model predicts the token with higher confidence than the reference model.

**Statistical Characteristics.** The analysis of over 260,000 token comparisons reveals that the loss difference distribution is governed by extreme events rather than Gaussian behavior. As shown in Figure 6(a), the data exhibits massive heavy tails. The excess kurtosis reaches 82.9 for members and 89.1 for non-members, far surpassing the zero value expected of a normal distribution. Furthermore, the distributions show significant positive skewness (7.5 for members and 7.6 for non-members). This statistical profile indicates that while the mean difference between members ($0.032\pm0.034$) and non-members ($0.007 \pm 0.029$) allows for some separability, the global average is highly susceptible to being skewed by rare, high-magnitude outliers.

**Domain vs. Membership Signals.** Figure 6(b) illustrates that the right tail regions (high loss reduction) for both members and non-members are nearly overlapping. This observation suggests that tokens exhibiting the most dramatic loss reductions are often poor indicators of membership. We hypothesize that these high-magnitude events correspond to *domain-specific tokens*—terms that appear frequently across the entire fine-tuning dataset (e.g., formatting keywords or common domain terminology). The model adapts to these features globally, resulting in improved reconstruction for both member and non-member instances drawn from the same domain.

To validate this hypothesis, we isolate tokens with the maximum mean loss differences and calculate their "Signal Strength," defined as the ratio of the average member difference to the average non-member difference. A ratio close to 1.0 implies the token provides no discriminative value. Table 12 presents several such tokens identified in the ArXiv dataset. Words like "Background," "Biography," and "Joseph" exhibit large mean differences ($> 0.39$), yet their signal strength is remarkably close to 1.0 (e.g., 0.973 for "Background"). These tokens contribute disproportionately to the global loss magnitude, effectively masking the subtler, instance-specific signals necessary for inference.

**Implications for Attack Design.** *The strongest membership signals are not found in the greatest loss reduction tokens, but rather in the consistent, smaller shifts of instance-specific tokens.* Standard attacks that rely on global averaging (like the Loss or Ratio baselines) are inherently noisy because they conflate these two signal types. By employing robust subset aggregation and sign-based voting,

Table 12: Tokens with the highest loss differences often yield weak membership signals. The Signal Strength ($\approx 1.0$) indicates that the loss reduction is similar for both members and non-members, confirming these are domain adaptation effects rather than instance-specific memorization.

| Token | Mean Diff ($\triangle$) | Signal Strength | Count |
|---|---|---|---|
| Biography | 1.040 | 0.705 | 50 |
| Background | 0.671 | 0.973 | 25 |
| Life | 0.631 | 0.701 | 21 |
| List | 0.544 | 0.501 | 39 |
| Description | 0.399 | 0.526 | 23 |
| Joseph | 0.393 | 0.978 | 23 |

SAMA effectively filters out the high-magnitude domain noise shown in Table 12 and amplifies the consistent, albeit weaker, signals of true membership.

