# OpenReview forum: "Membership Inference Attacks Against Fine-tuned Diffusion Language Models"
_ICLR.cc/2026/Conference — ICLR 2026 Poster_

### Official Review · Reviewer_9XsJ · 2025-10-29

**Soundness:** 3
**Presentation:** 3
**Contribution:** 2
**Rating:** 4
**Confidence:** 5

**Summary:**

The paper presents SAMA, a new Membership Inference Attack method tailored to Diffusion Language Models. The authors argue that the iterative masking mechanism in DLMs creates unique opportunities for membership probing and propose an attack that aggregates membership signals across progressively sampled masking configurations using robust sign-based statistics. Experiments on multiple datasets and DLM architectures show that SAMA outperforms prior MIA methods.

**Strengths:**

- The paper identifies an emerging gap by studying MIAs on DLMs, which are indeed less explored compared to autoregressive models.

- The formulation of the problem and discussion of bidirectional masking effects are insightful.

- The use of sign-based aggregation to handle heavy-tailed loss noise is conceptually sound.

**Weaknesses:**

- The attack assumes the adversary has grey-box access to both the fine-tuned DLM and its reference pre-trained model. In practice, access to the exact pre-trained model is often unrealistic, especially for proprietary commercial models. In Appendix D.5, we also see that performance drops significantly when the reference diverges. This raises doubts about the practical feasibility and real-world impact of the attack.

- Each attack step involves querying the target model with multiple masking configurations across many subset samples (T×N queries), which could easily require thousands of model queries per instance. This not only makes the attack computationally expensive but also highly detectable in monitored API environments. Standard defenses such as API rate limiting, randomization, or output obfuscation would likely render SAMA ineffective.

- The comparison with baselines seems unfair. SAMA requires orders of magnitude more queries than existing MIA approaches, yet the paper reports results without normalizing for query cost. The reported AUC gains may primarily reflect this increased probing power rather than a superior inference mechanism. A fair evaluation should constrain all methods to the same query budget or analyze performance vs. the number of queries.

- The work assumes Diffusion Language Models are widely deployed in practice. However, as of now, most production LLMs are still autoregressive (GPT, Claude, Gemini, etc.), while DLMs are largely at the research stage. Without stronger evidence that DLMs are being fine-tuned and deployed at scale, the urgency and real-world relevance of the attack remain limited.

**Questions:**

- How realistic is it for an adversary to obtain access to both the fine-tuned DLM and its corresponding pre-trained reference model, particularly for proprietary or commercial systems?


- What is the total number of queries required per target sample, and how does the attack’s performance scale with reduced query budgets?

- Could common API defenses such as rate limiting, randomized outputs, or noise injection effectively thwart SAMA?

- How much of SAMA’s reported advantage is attributable to increased query opportunities rather than algorithmic superiority?

- Can the authors provide evidence or examples of Diffusion Language Models being used in real-world production or fine-tuning scenarios?

---

> ### Author Response · Authors · 2025-11-26
>
> We thank Reviewer 9XsJ for the thoughtful review and are encouraged that the reviewer found our method "conceptually sound" and the problem formulation "insightful." Revisions made to the paper in response to the reviews are highlighted in blue for clarity and will be uploaded on November 29.

---

> ### Author Response · Authors · 2025-11-26
>
> ### W1 & Q1: Threat Model and Reference Model Access
>
> > (W1) "The attack assumes the adversary has grey-box access to both the fine-tuned DLM and its reference pre-trained model. In practice, access to the exact pre-trained model is often unrealistic, especially for proprietary commercial models."
> >
> > (Q1) "How realistic is it for an adversary to obtain access to both the fine-tuned DLM and its corresponding pre-trained reference model, particularly for proprietary or commercial systems?"
>
> Thank you. Our threat model is relevant to both current open-weight deployment and future API designs for Diffusion Language Models (DLMs).
>
> **(1)** **API access proposals:**  Since DLMs are quite new, there is not yet a well-stablished API for them.  One of the promising API proposals *In-filling demo* for LLaDA [1] provides a "custom partially masked sequence" (the text with a blank), which provides the information needed in SAMA. Also, widely used serving frameworks for ARMs, such as vLLM, support parameters like `prompt_logprobs` to return token-level probabilities for input sequences [2].
>
> **(2)** **Open-Weight Models:** Models like our targets (LLaDA, Dream) are open-weight. Platforms like HuggingFace host thousands of fine-tuned models alongside their original base versions. In this scenario, an organization fine-tunes a publicly available base model on private data and releases the weights (e.g., `tenzro/daml-coder`). The attacker therefore has direct access to both the target (fine-tuned) and the reference (base) models.
>
> Finally, SAMA remains robust when the exact reference is unavailable. As shown in Figure 3 (Appendix D.5), when we use a misaligned reference (using `Dream-v0`, which shares the same scale with Qwen 2.5-7B, as the reference model to attack `LLaDA`), SAMA still achieves an AUC of \~0.73. In contrast, standard baselines drop to near-random performance (~0.50-0.60) under the same conditions. This indicates that while a perfect reference helps, SAMA’s sign-based aggregation extracts a meaningful signal even from a suboptimal reference.
>
> [1]
> multimodalart. (n.d.). *LLaDA*. Hugging Face. https://huggingface.co/spaces/multimodalart/LLaDA
>
> [2] vLLM. (n.d.). *LogprobsProcessor — prompt_logprobs*. vLLM documentation. https://docs.vllm.ai/en/latest/api/vllm/v1/engine/logprobs/#vllm.v1.engine.logprobs.LogprobsProcessor.prompt_logprobs

---

> ### Author Response · Authors · 2025-11-26
>
> ### W2, W3, Q2, & Q4: Query Cost & Fair Comparison
>
> > (W2) "Each attack step involves querying the target model with multiple masking configurations across many subset samples (T×N queries), which could easily require thousands of model queries per instance. This not only makes the attack computationally expensive but also highly detectable in monitored API environments."
> >
> > (W3) "The comparison with baselines seems unfair. SAMA requires orders of magnitude more queries than existing MIA approaches, yet the paper reports results without normalizing for query cost. The reported AUC gains may primarily reflect this increased probing power rather than a superior inference mechanism."
> >
> > (Q2) "What is the total number of queries required per target sample, and how does the attack’s performance scale with reduced query budgets?"
> >
> > (Q4) "How much of SAMA’s reported advantage is attributable to increased query opportunities rather than algorithmic superiority?"
>
>
> We appreciate the opportunity to clarify this key point. The concern regarding high query complexity stems from a misunderstanding of our algorithm: **SAMA requires only $T$ model queries in total, which is the same as all query-based baselines.**
>
> **1. Clarification of Query Volume ($T$ vs $T \times N$)**
> In each of the $T$ diffusion queries, one provides as input a token sequence where $k_t$ tokens are masked, and the output includes the probability distributions for all $k_t$ masked tokens.  SAMA then chooses $N$ random subsets of the $k_t$ masked tokens, and for each such subset aggregates the information for the tokens in the subset. <!--The sampling of $N=128$ subsets happens entirely **offline on the CPU** using these returned probabilities.--> This aggregation step is computationally trivial (<1% of total wall-clock time) compared to the model inference.
>
> **2. Fair Comparison (Identical Budgets)**
> To ensure fairness, we use the same query cost in experimental comparisons of SAMA versus other baseline attacks.  All attacks use $T$ DLM queries for each instance.  <!--We **fixed the model query budget to $T=16$ for SAMA *and all* query-based baselines**. Therefore, the computational bottleneck is identical for SAMA and the baselines it outperforms. -->
>
> Here we present the actual runtime logs for the LLaDA-8B-Base model. SAMA’s runtime is nearly identical to the `Ratio` baseline:
>
> | Dataset | Loss | Ratio | **SAMA (Ours)** | Neighborhood |
> | :--- | :--- | :--- | :--- | :--- |
> | **Wikipedia (en)** | 39m 01s | 1h 32m 00s | **1h 32m 16s** | 8h 43m 27s |
> | **PubMed Central** | 55m 17s | 1h 37m 04s | **1h 36m 19s** | 9h 32m 38s |
>
> **3. Robustness at Low Query Budgets ($T=4$)**
> Finally, we also evaluated SAMA with a restricted budget of just $T=4$ queries. As shown below, SAMA maintains high attack performance even with fewer queries, outperforming full-budget baselines.
>
> | Dataset | Loss ($T=16$) | Ratio ($T=16$) | **SAMA ($T=4$)** |
> | :--- | :--- | :--- | :--- |
> | **TPR @ 1% FPR** | 0.052 | 0.055 | **0.257** |
> | **Runtime** | 55m 17s | 1h 37m 04s | **23m 05s** |
>
> We will include these runtime comparisons and the low-budget ablation study in Appendix D.7 to clearly demonstrate SAMA's efficiency.

---

> ### Author Response · Authors · 2025-11-26
>
> ### Q3: On API Defenses
>
> > "Could common API defenses such as rate limiting, randomized outputs, or noise injection effectively thwart SAMA?"
>
> * **Rate Limiting:** Since SAMA requires only **$T$ queries** per sample, and we have shown that setting $T$ to be as small as $4$ remains effective, SAMA is  a very low-volume attack, and rate limiting is not an effective defense. <!--ineffe it is no more vulnerable to rate limiting than *any* baseline method that uses a standard number of Monte Carlo samples for a stable estimate. It would not be flagged as a high-volume attack. -->
> * **Output Randomization & Noise Injection:** These defenses face a fundamental trade-off: adding enough noise to mask membership signals often degrades the utility of the model’s generation.

---

> ### Author Response · Authors · 2025-11-26
>
> ### W4 & Q5: Real-World Relevance of DLMs
>
> > (W4) "The work assumes Diffusion Language Models are widely deployed in practice... the urgency and real-world relevance of the attack remain limited."
> >
> > (Q5) "Can the authors provide evidence or examples of Diffusion Language Models being used in real-world production...?"
>
> While Autoregressive Models currently dominate production, DLMs are a rapidly advancing competitive alternative. This is evidenced by recent high-performance models like LLaDA and Dream we used in our evaluation, and major research efforts like Google's recent work on "Gemini Diffusion" [1]. Gemini Diffusion's external benchmark performance is comparable to much larger models, whilst also being faster.
>
> Security research plays a vital role in anticipating vulnerabilities *before* a technology is widely deployed with sensitive data. Our findings are timely for practitioners designing the DLM API standard; specifically, our work highlights that APIs for these models must be designed carefully (e.g., restricting access to full log-probabilities for masked spans) to prevent the specific leakage paths we have identified.
>
> [1] DeepMind. (n.d.). Gemini Diffusion. DeepMind. https://deepmind.google/models/gemini-diffusion/

---

### Official Review · Reviewer_7q5n · 2025-10-31

**Soundness:** 3
**Presentation:** 3
**Contribution:** 2
**Rating:** 6
**Confidence:** 3

**Summary:**

This paper introduces SAMA (Subset-Aggregated Membership Attack), a novel membership inference attack tailored for diffusion-based language models (DLMs) in fine-tuning scenarios. Unlike traditional autoregressive models, DLMs generate text through iterative bidirectional masking, which exposes unique privacy vulnerabilities. SAMA exploits these characteristics by employing progressive masking and sign-based subset aggregation to amplify sparse membership signals that emerge under specific mask configurations. Through extensive experiments across multiple datasets and diffusion architectures, the authors show that SAMA achieves substantial improvements over established baselines (previously mainly designed for autoregressive models), such as loss-based and ZLIB attacks, particularly at low false-positive rates.

**Strengths:**

1. The paper focuses on a new problem by systematically investigating membership inference vulnerabilities in diffusion-based language models, a model family that differs fundamentally from autoregressive architectures.


2. The proposed SAMA framework combines progressive masking and sign-based subset aggregation to effectively expose membership vulnerabilities.


3. The experimental evaluation is extensive and well-controlled, spanning multiple datasets, and various MIA baselines.


4. The ablation analysis is informative, isolating the contribution of each component in SAMA.

5. The paper is well-written and easy to follow.

**Weaknesses:**

1. The proposed method is specialized for diffusion-based language models and depends on access to compatible reference models, limiting its generalizability to broader LLM architectures.


2. The intuition behind progressive masking and subset aggregation remains somewhat abstract; a concrete example with simple proof or visualization would make the mechanism more interpretable.


3. The method incurs high query complexity due to multiple mask configurations, raising scalability concerns for large models and long sequences (the $m$ can be large).


4. The paper focuses primarily on empirical MIA results. Better to have more in-depth discussion on the concrete patterns or examples in the evaluation.


5. It is better to discuss the potential defenses strategies for SAMA.

**Questions:**

1. Could SAMA be generalized beyond diffusion-based architectures, for example, to masked or autoregressive language models, by adapting the masking or sampling strategy?


2. What is the computational overhead of SAMA in practice? Since it involves repeated masking and querying, quantifying its runtime or query cost relative to traditional MIAs would help assess its feasibility in realistic attack settings.

---

> ### Author Response · Authors · 2025-11-26
>
> We thank Reviewer 7q5n for their positive feedback on our work, particularly for acknowledging "effectively expose membership vulnerabilities" and "experimental evaluation is extensive and well-controlled." Revisions made to the paper in response to the reviews are highlighted in blue for clarity and will be uploaded on November 29.

---

> ### Author Response · Authors · 2025-11-26
>
> ### W1 & Q1: Generalizability and Specialization
> > (W1) "The proposed method is specialized for diffusion-based language models and depends on access to compatible reference models, limiting its generalizability to broader LLM architectures."
> >
> > (Q1)"Could SAMA be generalized beyond diffusion-based architectures, for example, to masked or autoregressive language models, by adapting the masking or sampling strategy?"
>
> We appreciate this insightful question regarding the scope of our method. SAMA is specialized precisely because DLMs present a fundamentally different attack surface than ARMs. As our results demonstrate, existing attacks designed for ARMs fail on DLMs (yielding near-random performance), which necessitates a tailored approach to expose these new vulnerabilities.
>
> Regarding generalization, SAMA relies on two mechanisms at a high level: progressive masking across configurations and robust aggregation within configurations. While progressive masking is specific to the stochastic nature of DLMs and does not apply to the deterministic, left-to-right structure of ARMs, the robust subset aggregation component, however, is highly generalizable. By using sign-based voting to filter out heavy-tailed noise, this technique could likely enhance attacks on ARMs or masked language models that face similar signal-to-noise challenges. We will explicitly add this distinction to our discussion to clarify where SAMA's principles can extend beyond diffusion models.

---

> ### Author Response · Authors · 2025-11-26
>
> ### W2 & W4: Intuition and Concrete Examples
>
> > "The intuition behind progressive masking and subset aggregation remains somewhat abstract; a concrete example with simple proof or visualization would make the mechanism more interpretable."
> > "The paper focuses primarily on empirical MIA results. Better to have more in-depth discussion on the concrete patterns or examples in the evaluation."
>
> These are excellent suggestions. We will make two additions to the paper to improve interpretability.
>
> * **Illustrative Plot (W2):** To make the mechanism less abstract, **we will add a new illustrative plot** (and reference it in Sec 3.1). This plot will show the distribution of signal strengths ($\Delta_{DF}$) from 100+ random mask configurations for a single member and non-member sample. This will visually confirm our core hypothesis: for members, the signal is sparse and highly varied, while for non-members, it's centered at zero, demonstrating *why* simple averaging fails and robust aggregation is necessary.
>
> * **Qualitative Examples (W4):** To provide a more "in-depth discussion" beyond metrics, **we will add a new section to the Appendix E with qualitative examples** of high-scoring (member) and low-scoring (non-member) samples from our evaluation (e.g., from the GitHub and ArXiv datasets). This will provide a more concrete view of the memorized patterns SAMA successfully identifies.

---

> ### Author Response · Authors · 2025-11-26
>
> ### W3 & Q2: Computational Cost and Scalability
> > (W3) "The method incurs high query complexity due to multiple mask configurations, raising scalability concerns for large models and long sequences (the m can be large)"
> > (Q2) "What is the computational overhead of SAMA in practice? Since it involves repeated masking and querying, quantifying its runtime or query cost relative to traditional MIAs would help assess its feasibility in realistic attack settings."
>
> We thank the reviewer for raising this, as it highlights a critical point of our experimental design that we must make clearer.
>
> The reviewer's concern about high query complexity is based on a misunderstanding that SAMA's $T$ steps are *additional* cost. This is not the case; we used a **fixed-budget comparison**.
>
> A single-query loss on a DLM is an unreliable signal, as the loss itself is an expectation over random masks. Any stable loss-based attack (baseline or not) *must* use Monte Carlo sampling. To ensure a **fair comparison**, we **fixed the model query budget to T=16 for SAMA *and all* query-based baselines** (e.g., Loss, Ratio, SecMI, PIA). Therefore, the primary computational bottleneck—the 16 forward passes—is **identical for SAMA and the baselines it outperforms**. SAMA's only different cost is the CPU-side subset aggregation, which is computationally trivial (e.g., <1% of total wall-clock time) compared to the 8B-parameter model inference. As shown in the table below, in practice, SAMA’s runtime is nearly identical to the Ratio baseline (the second-best performing method) on the LLaDA-8B-Base model.
>
> | Dataset | Loss | Ratio | **SAMA (Ours)** | Neighborhood |
> | :--- | :--- | :--- | :--- | :--- |
> | **Wikipedia (en)** | 39m 01s | 1h 32m 00s | **1h 32m 16s** | 8h 43m 27s |
> | **PubMed Central** | 55m 17s | 1h 37m 04s | **1h 36m 19s** | 9h 32m 38s |
> | **HackerNews** | 54m 20s | 1h 35m 08s | **1h 35m 36s** | 9h 13m 02s |
>
> We will **explicitly state this fixed-budget comparison** in Section 4.1. We will also include the complete theoretical analysis and empirical comparisons of the computational cost of SAMA versus the baselines as requested in our revision's Appendix D.7.

---

> ### Author Response · Authors · 2025-11-26
>
> ### W5: Discussing Potential Defenses
> > "It is better to discuss the potential defenses strategies for SAMA."
>
> Thank you for the suggestion. Our paper provides a detailed analysis of LoRA and DP-LORA as defenses in Appendix D.5. We will add a state-of-the-art **training-time defense** (SOFT [1]) and an **inference-time defense** (Temperature scaling) as D.5.3 and D.5.4 in our revision.
>
> ---
> [1] Zhang et al. "SOFT: Selective Data Obfuscation for Protecting LLM Fine-tuning against Membership Inference Attacks." USENIX Security 2025.

---

### Official Review · Reviewer_VbcU · 2025-10-31

**Soundness:** 4
**Presentation:** 3
**Contribution:** 3
**Rating:** 6
**Confidence:** 4

**Summary:**

This paper presents the first systematic investigation into the vulnerability of fine-tuned Diffusion Language Models (DLMs) to Membership Inference Attacks (MIAs). The authors identify that the bidirectional, multi-configuration masking mechanism of DLMs creates a fundamentally different and larger attack surface compared to autoregressive models (ARMs). To exploit this, they propose a novel attack framework named SAMA (Subset-Aggregated Membership Attack). SAMA leverages a progressive masking strategy to probe the model at multiple scales, combined with a robust subset aggregation method using sign-based statistics to handle sparse and noisy membership signals. Through extensive experiments on two state-of-the-art DLMs across nine datasets, the authors demonstrate that SAMA significantly outperforms a wide range of existing MIA baselines, revealing critical and previously overlooked privacy risks in this emerging class of models.

**Strengths:**

-  The work is highly significant and original as it provides the first systematic study of Membership Inference Attacks against the emerging and important class of Diffusion Language Models. It tackles a critical and underexplored privacy problem.
-  The quality of the work is high. The proposed SAMA attack is technically sound and specifically tailored to the unique properties of DLMs. The empirical evaluation is comprehensive, and the substantial performance gains over a wide range of baselines strongly support the paper's claims.
-   The paper is written with clarity. The core concepts, especially the insightful comparison between the attack surfaces of ARMs and DLMs, are well-articulated, making the motivation for the proposed method easy to understand.

**Weaknesses:**

- The grey-box access model assumed in Section 2.1 appears to be quite strong. It posits that the adversary can query the target model with arbitrary, custom partially masked sequences and receive detailed outputs like logits for specific token positions. This may not be realistic in many practical scenarios. For instance, with closed-source models, an attacker might only be able to access the final output sequence and its token probabilities via an API. Conversely, with open-source models, an attacker would have full white-box access, including parameters and gradients, which is an even stronger (and different) threat model. The assumed threat model sits in a potentially unrealistic middle ground. The authors should provide a stronger justification for its practicality or discuss how the attack's performance might change under more constrained, API-based settings.
-   While the components of SAMA are intuitive and empirically effective, their design is largely heuristic and lacks theoretical guarantees. For example, the analogy drawn between progressive masking and multi-scale analysis techniques like wavelet decomposition (line 291) is an interesting intuition but is not rigorously justified; the mathematical mechanisms and application domains are fundamentally different.
-  A significant concern after reading the methodology is the computational overhead of SAMA. A standard loss-based attack on an ARM requires roughly one forward pass per sample (complexity $O(L)$). In contrast, the proposed SAMA algorithm requires multiple forward passe, e.g. one for each of the $T$ progressive masking steps. This suggests a complexity that is at least $T$ times higher. While the impressive performance gains might justify this extra cost, the authors should discuss this directly. A theoretical analysis and an empirical comparison of the runtime/computational cost of SAMA versus the baselines are essential for evaluating the practical viability of the attack.

**Questions:**

1. The grey-box assumption in Section 2.1 is my primary concern. Could the authors clarify if the cited works for this threat model (Zhai et al., 2024; Duan et al., 2023; etc.) truly operate under this specific assumption for language models (i.e., querying with custom masks to get intermediate losses)? Or is this a stronger variant? How would the attack perform in a more realistic API-based scenario where only the final generated sequence and its token probabilities are available?
2. Section 3.1 provides an intuitive analysis that multiple masking configurations in DLMs create a larger attack surface. While Section 3.2 points to the main results in Section 4.2 for empirical validation, is there any more direct experimental data that can specifically isolate and demonstrate how varying mask configurations ($S$) impact the membership signal strength ($\Delta_{DF}(x;S)$)?
3. The attack seems to rely on sampling a variety of mask configurations with the hope of "hitting" one that reveals a strong signal. Is there an analysis of how many samples are needed to achieve good performance? Specifically, how does the attack's performance scale with the number of progressive steps ($T$) and the number of sampled subsets ($N$)?
4. The subset aggregation method relies on hyperparameters $N$ (number of subsets) and $m$ (subset size). How were the values ($N=128, m=10$) chosen? Could the authors provide a sensitivity analysis showing how these parameters affect the trade-off between attack accuracy and computational cost?
5. The analysis in Section 3.1 is compelling, but it is presented without direct empirical support within the section itself. Could the authors include a small, illustrative experiment to support the claim that different mask configurations can reveal stronger or weaker signals for member vs. non-member samples?
6. A minor point on presentation: the manuscript uses a large number of em-dashes for sentence construction, which can hinder readability. I would recommend the authors polish the text to improve flow and clarity, which would help lower the cognitive load for readers.

---

> ### Author Response · Authors · 2025-11-26
>
> We thank Reviewer VbcU for saying our work is "highly significant and original", that the "quality of the work is high", and that "paper is written with clarity". We will revise the manuscript to clarify the questions. Revisions made to the paper in response to the reviews are highlighted in blue for clarity and will be uploaded on November 29.

---

> ### Author Response · Authors · 2025-11-26
>
> ### W1 Practicality of the Grey-Box Threat Model
>
> > (W1) "The grey-box access model assumed in Section 2.1 appears to be quite strong. [...] The authors should provide a stronger justification for its practicality or discuss how the attack's performance might change under more constrained, API-based settings."
>
> Thank you. Our revision will provide further justification for the **grey-box access model**'s practicality by connecting it to plausible deployment scenarios for Diffusion Language Models (DLMs). Our threat model is relevant to both current open-weight deployments and future API designs for DLMs.
>
> Since DLMs are quite new, there is not yet a well-established API for them.  One of the promising proposals *In-filling demo* for LLaDA [1] provides a "custom partially masked sequence" (the text with a blank), which provides the information needed in SAMA. Furthermore, widely used serving frameworks for ARMs, such as vLLM, support parameters like `prompt_logprobs` to return token-level probabilities for input sequences [2].  It is entirely plausible that similar production-serving frameworks designed for DLMs will provide this same level of access to allow users to effectively debug, control, or audit the bidirectional prediction mechanism.
>
> Moreover, the assumption of access to per-token probabilities (or loss) is a standard convention in the literature of membership inference attacks on LLMs, regardless of architecture. State-of-the-art attacks on LLMs, such as Min-K% and Min-K%++ [3-4], also rely on accessing the probabilities of specific tokens to distinguish members from non-members.
>
> We will revise Section 2.1 to explicitly connect our assumption to these **in-filling APIs** and the **established norms of MIA evaluation**.
>
> ---
> [1] multimodalart. (n.d.). *LLaDA*. Hugging Face. https://huggingface.co/spaces/multimodalart/LLaDA
>
> [2] vLLM. (n.d.). *LogprobsProcessor — prompt_logprobs*. vLLM documentation. https://docs.vllm.ai/en/latest/api/vllm/v1/engine/logprobs/#vllm.v1.engine.logprobs.LogprobsProcessor.prompt_logprobs
>
> [3] Shi, Weijia, et al. "Detecting pretraining data from large language models." arXiv preprint arXiv:2310.16789 (2023).
>
> [4] Zhang, Jingyang, et al. "Min-k%++: Improved baseline for detecting pre-training data from large language models." arXiv preprint arXiv:2404.02936 (2024).

---

> ### Author Response · Authors · 2025-11-26
>
> ### W2: Heuristic Design
>
> > "While the components of SAMA are intuitive and empirically effective, their design is largely heuristic and lacks theoretical guarantees. For example, the analogy drawn between progressive masking and multi-scale analysis techniques like wavelet decomposition (line 291) is an interesting intuition but is not rigorously justified; the mathematical mechanisms and application domains are fundamentally different."
>
>
> We appreciate the opportunity to clarify the theoretical grounding of our design choices. The wavelet analogy (line 291) was served more as intuition than rigorous justification, and we will revise this framing in the final version. That said, SAMA's design is grounded in principled reasoning from robust statistics, which we now elaborate.
>
> Intuitively, while large loss reductions (between target and base models) tend to indicate membership, our added experiments in Appendix E on real data show this is an oversimplification: Large loss reduction in tokens can be achieved by tokens in the domain of the training data, but not necessarily member tokens. Fine-tuning on a dataset will have domain tokens appear frequently during fine-tuning. But these domain-specific tokens are poor signals for membership of any single instance. Thankfully, such tokens happen infrequently. On the other hand, instance-specific tokens, which are those unique to the individual instance and rare in the general domain, typically exhibit smaller loss reductions. However, these are the true membership signals. Unlike domain tokens, which cause sharp, isolated drops in loss, these sample-specific tokens show a subtle but consistent improvement across the sequence, which our sign-based aggregation is designed to capture.
>
> The key insight is that LLM MIA needs to distinguish a **true membership signal** (weak, consistent) from the noise of **domain adaptation** (strong, sparse). SAMA's components are designed to first *isolate* this weak signal and then *amplify* it.
>
> **Why robust subset aggregation**
>
> This is the core innovation of SAMA: the common averaging attack (like the `Loss` baseline) fails because the high-magnitude **Domain Signal** on just a few tokens (e.g., `import` in GitHub, or technical terms in ArXiv) tends to dominate and "drown out" the small, consistent **Membership Signals** that are spread across the whole sequence. As our ablation study (Figure 2) confirms, our robust aggregation is the most critical component, delivering the largest (20-30%) AUC gain.
>
> Appendix E presents a new token-level analysis that confirms this result. The loss-difference distribution has extremely heavy tails (kurtosis > 80), and the tokens showing the largest loss reductions are not the main contributors to membership signals; rather, they are strongly associated with domain-specific topics.
>
> Our **Robust Subset Aggregation** is a two-part strategy to solve this:
> 1.  **Local Subsetting:** Instead of one average over all masked tokens, we sample *many* small, random token subsets. The strong, sparse Domain Signal will only appear in a *fraction* of these subsets, limiting its influence. The consistent Membership Signal, however, will be present in *most* subsets.
> 2.  **Sign-Based Voting:** We then discard the noisy magnitudes entirely and use a **sign-based statistic**—a "vote" on whether the target loss is lower than the reference loss ($1[\Delta_{DF}^{n}(x;\mathcal{S}_{t})>0]$). This allows the *consistent* (but weak) Membership Signal to "outvote" the *few* (but strong) Domain Signals.
>
> **Why progressive masking**
>
> While subset aggregation *finds* the signal, progressive masking *amplifies* it. Unlike ARMs (which offer one fixed context), DLMs uniquely allow us to probe the model with an exponential number of masking configurations. We exploit this by systematically sampling masks at **progressively increasing densities** (e.g., 5% to 50%). This addresses a key trade-off we discussed in line 286:
> * **Sparse Masks** (low density) provide *cleaner, stronger per-token signals* due to more available context, but offer fewer data points to aggregate.
> * **Dense Masks** (high density) provide *more aggregation points* (more "votes"), but the individual signals are *noisier* as less context is available.
>
> **Why adaptive weighting**
>
> This component simply combines the evidence from the progressive masking steps. Since sparse masks provide cleaner signals, **Adaptive Weighting** strategically gives more weight to the "votes" from these earlier, sparser steps using inverse-step weighting. This prioritizes the most reliable signals while still incorporating evidence from all scales.

---

> ### Author Response · Authors · 2025-11-26
>
> ### W3: Computational Overhead of SAMA
>
> > "A significant concern after reading the methodology is the computational overhead of SAMA. A standard loss-based attack on an ARM requires roughly one forward pass per sample (complexity O(L)). In contrast, the proposed SAMA algorithm requires multiple forward passe, e.g. one for each of the T progressive masking steps. This suggests a complexity that is at least T times higher. While the impressive performance gains might justify this extra cost, the authors should discuss this directly. A theoretical analysis and an empirical comparison of the runtime/computational cost of SAMA versus the baselines are essential for evaluating the practical viability of the attack."
>
> We appreciate the reviewer’s focus on the practical viability of our method. We would like to clarify the computational cost from the following perspectives.
>
>
> **SAMA v.s. ARM Baselines**
> Indeed, SAMA performs $T$ independent queries (where $T=16$ in our evaluation) against a DLM.  However, the cost of one ARM query differs from that of one DLM query, as validated by previous works [1].  Our logs on the Wikipedia dataset show that the attack on the LLaDA-8B DLM required approximately 92 minutes, while the ratio attack on ARM Llama-3.1-8B ARM required 36 minutes.
>
> **SAMA v.s. DLM Baselines**
> Calculating the loss for a DLM is fundamentally different from an ARM. An ARM computes the loss for the entire sequence in a single pass. In contrast, a single DLM pass only evaluates one specific random mask pattern. Because the search space of masks is exponential, the loss from a single pattern is noisy and not representative of the sequence. To obtain a reliable signal, one must estimate the expected loss over multiple configurations (Monte Carlo estimation), as formalized in Equation 4. Therefore, ***any* query-based MIA inherently requires multiple passes when adapted to the DLM paradigm**. As shown in the table below, in practice, SAMA’s runtime is nearly identical to the Ratio baseline (the second-best performing method) on the LLaDA-8B-Base model.
>
> | Dataset | Loss | Ratio | **SAMA (Ours)** | Neighborhood |
> | :--- | :--- | :--- | :--- | :--- |
> | **Wikipedia (en)** | 39m 01s | 1h 32m 00s | **1h 32m 16s** | 8h 43m 27s |
> | **PubMed Central** | 55m 17s | 1h 37m 04s | **1h 36m 19s** | 9h 32m 38s |
> | **HackerNews** | 54m 20s | 1h 35m 08s | **1h 35m 36s** | 9h 13m 02s |
>
> **Robustness at Low Query Budgets**
> Finally, the number of steps ($T$) is a tunable hyperparameter. Our new ablation studies in Appendix D.4 indicate that when restricting the budget to just $T=4$ steps, SAMA maintains high attack performance with a TPR@1%FPR of **0.131 ± 0.03**, compared to **0.178 ± 0.03** in the $T=16$ setting. We will include the complete theoretical analysis and empirical comparisons of the computational cost of SAMA versus the baselines as requested in our revision's Appendix D.7.
>
> | Dataset | Loss | Ratio | **SAMA (Ours T=4)** | Neighborhood |
> | :--- | :--- | :--- | :--- | :--- |
> | **Wikipedia (en)** | 10m 03s | 24m 03s | **23m 4s** | 8h 24m 03s |
> | **PubMed Central** | 12m 46s | 23m 04s | **23m 5s** | 8h 57m 34s |
> | **HackerNews** | 13m 42s | 23m 08s | **23m 54s** | 8h 43m 02s |
>
>
> [1] Ye, Jiacheng, et al. "Dream 7b: Diffusion large language models." arXiv preprint arXiv:2508.15487 (2025).

---

> ### Author Response · Authors · 2025-11-26
>
> ### Q1: Clarification of the Grey-Box Threat Model
> > "Could the authors clarify if the cited works for this threat model (Zhai et al., 2024; Duan et al., 2023; etc.) truly operate under this specific assumption for language models (i.e., querying with custom masks to get intermediate losses)? Or is this a stronger variant? How would the attack perform in a more realistic API-based scenario where only the final generated sequence and its token probabilities are available?"
>
> The works [1-2] operate under the same assumption in vision tasks: querying with custom perturbations (noise or masks) to measure reconstruction fidelity. While our requirement for token probabilities is technically "stronger" than observing pixels, it is the necessary equivalent for calculating reconstruction loss in the discrete language domain. We will clarify this. Regarding the restricted API scenario: SAMA cannot be applied if limited to the final sequence (i.e. a "label-only senario"). However, as detailed in W1, such masking interfaces are consistent with emerging DLM deployment approaches.
>
> ---
> [1] Duan, Jinhao, et al. "Are diffusion models vulnerable to membership inference attacks?." International Conference on Machine Learning. PMLR, 2023.
>
> [2] Matsumoto, Tomoya, Takayuki Miura, and Naoto Yanai. "Membership inference attacks against diffusion models." 2023 IEEE Security and Privacy Workshops (SPW). IEEE, 2023.

---

> ### Author Response · Authors · 2025-11-26
>
> ### Q2 & Q5: Direct Evidence for Signal Sparsity
>
> > (Q2) "Section 3.1 provides an intuitive analysis that multiple masking configurations in DLMs create a larger attack surface. While Section 3.2 points to the main results in Section 4.2 for empirical validation, is there any more direct experimental data that can specifically isolate and demonstrate how varying mask configurations (S) impact the membership signal strength $\Delta_{DF}(x;S)$?"
> >
> > (Q5) "The analysis in Section 3.1 is compelling, but it is presented without direct empirical support within the section itself. Could the authors include a small, illustrative experiment to support the claim that different mask configurations can reveal stronger or weaker signals for member vs. non-member samples?"
>
> These are excellent suggestions. A direct visualization will strongly motivate our design.
>
> **Revision Plan:** We will **add a new illustrative experiment and plot to Section 3.1.**
>
> This plot will sample from both the member and a non-member set. For a *fixed* masking density (e.g., $\alpha=20\%$), we will sample 100+ different random mask configurations ($S_1, S_2, ... S_{100}$) and plot the *full distribution* of their resulting signal strengths ($\Delta_{DF}(x;S)$).
>
> This visualization will directly support our core hypothesis  by showing:
> 1.  The non-member's distribution will be a tight, symmetric Gaussian centered at zero (pure noise).
> 2.  The member's distribution will be shifted positive, broad, and heavy-tailed, visually demonstrating that most masks give a weak/noisy signal while a few "hit" and provide a strong positive signal.
>
> This plot will provide clear, empirical justification for *why* a simple averaging attack fails and *why* SAMA's robust, sign-based aggregation is necessary.

---

> ### Author Response · Authors · 2025-11-26
>
> ### Q3 & Q4: Hyperparameter Sensitivity Analysis
>
> > (Q3) "The attack seems to rely on sampling a variety of mask configurations with the hope of "hitting" one that reveals a strong signal. Is there an analysis of how many samples are needed to achieve good performance? Specifically, how does the attack's performance scale with the number of progressive steps (T) and the number of sampled subsets (N)?"
> >
> > (Q4) "The subset aggregation method relies on hyperparameters N (number of subsets) and m (subset size). How were the values (N=128, m=10) chosen? Could the authors provide a sensitivity analysis showing how these parameters affect the trade-off between attack accuracy and computational cost?"
>
> We selected the hyperparameters ($N=128, m=10, T=16$) **empirically** to achieve a balance between attack performance and computational cost.
>
> To address your request, we will conduct a sensitivity analysis and include it in Appendix D.4. Specifically, we will run experiments varying:
> * **The number of subsets ($N$) and progressive steps ($T$):** To demonstrate how the attack's performance scales with computational budget.
> * **The subset size ($m$):** To show the impact of subset granularity on the robust aggregation.
>
> We will provide plots visualizing these trade-offs to justify our chosen configuration.

---

### Official Review · Reviewer_aFkU · 2025-11-01

**Soundness:** 3
**Presentation:** 3
**Contribution:** 3
**Rating:** 6
**Confidence:** 3

**Summary:**

This paper presents the first systematic investigation of Membership Inference Attack (MIA) vulnerabilities in fine-tuned Diffusion Language Models. It introduces SAMA, a novel attack framework that leverages progressive masking and robust sign-based aggregation to exploit the bidirectional dependencies unique to DLMs. The method demonstrates significant improvements over a wide range of baselines across multiple models and datasets.

**Strengths:**

1. The proposed SAMA method is innovative since it is the first work to systematically study MIA risks for Diffusion Language Models.
2. The experimental setup is comprehensive, evaluating SAMA on state-of-the-art DLMs across diverse datasets against multiple baselines. The results are compelling and the ablation studies effectively justify the design choices.
3. The paper is written in a clear structure.

**Weaknesses:**

1. The SAMA framework involves multiple components (e.g., progressive masking, subset aggregation, adaptive weighting) and hyperparameters. While effective, the method is somewhat complex. A more intuitive explanation and stronger justification for the necessity of each component could enhance clarity.
2. It would be good to propose or evaluate more defensive strategies to improve practical applicability.

**Questions:**

See the weaknesses.

---

> ### Author Response · Authors · 2025-11-26
>
> We thank the reviewer for their feedback and constructive suggestions. We are encouraged that the reviewer found our work "innovative," our "experimental setup is comprehensive," and the paper's "clear structure." Revisions made to the paper in response to the reviews are highlighted in blue for clarity and will be uploaded on November 29.

---

> ### Author Response · Authors · 2025-11-26
>
> ### W1: Intuition and Justification for SAMA's Components
>
> > "[...] A more intuitive explanation and stronger justification for the necessity of each component could enhance clarity."
>
> We appreciate this suggestion and will clarify the intuition behind SAMA's design.
>
> Intuitively, while large loss reductions (between target and base models) tend to indicate membership, our added experiments in Appendix E on real data show this is an oversimplification: Large loss reduction in tokens can be achieved by tokens in the domain of the training data, but not necessarily member tokens. Fine-tuning on a dataset will have domain tokens appear frequently during fine-tuning. But these domain-specific tokens are poor signals for membership in any single instance. Thankfully, such tokens happen infrequently. On the other hand, instance-specific tokens, which are those unique to the individual instance and rare in the general domain, typically exhibit smaller loss reductions. However, these are the true membership signals. Unlike domain tokens, which cause sharp, isolated drops in loss, these sample-specific tokens show a subtle but consistent improvement across the sequence, which our sign-based aggregation is designed to capture.
>
> The key insight is that LLM MIA needs to distinguish a **true membership signal** (weak, consistent) from the noise of **domain adaptation** (strong, sparse). SAMA's components are designed to first *isolate* this weak signal and then *amplify* it.
>
> **Why robust subset aggregation**
>
> This is the core innovation of SAMA: the common averaging attack (like the `Loss` baseline) fails because the high-magnitude **Domain Signal** on just a few tokens (e.g., `import` in GitHub, or technical terms in ArXiv) tends to dominate and "drown out" the small, consistent **Membership Signals** that are spread across the whole sequence. As our ablation study (Figure 2) confirms, our robust aggregation is the most critical component, delivering the largest (20-30%) AUC gain.
>
> Appendix E further validates this. The loss-difference distribution has long tails (kurtosis > 80), and the tokens showing the largest loss reductions are not the main contributors to membership signals.
>
> Our **Robust Subset Aggregation** is a two-part strategy to solve this:
> 1.  **Local Subsetting:** Instead of one average over all masked tokens, we sample *many* small, random token subsets. The strong, sparse Domain Signal will only appear in a *fraction* of these subsets, limiting its influence. The consistent Membership Signal, however, will be present in *most* subsets.
> 2.  **Sign-Based Voting:** We then discard the noisy magnitudes entirely and use a **sign-based statistic**—a "vote" on whether the target loss is lower than the reference loss ($1[\Delta_{DF}^{n}(x;\mathcal{S}_{t})>0]$). This allows the *consistent* (but weak) Membership Signal to "outvote" the *few* (but strong) Domain Signals.
>
> **Why progressive masking**
>
> While subset aggregation *finds* the signal, progressive masking *amplifies* it. Unlike ARMs (which offer one fixed context), DLMs uniquely allow us to probe the model with an exponential number of masking configurations. We exploit this by systematically sampling masks at **progressively increasing densities** (e.g., 5% to 50%). This addresses a key trade-off we discussed in line 286:
> * **Sparse Masks** (low density) provide *cleaner, stronger per-token signals* due to more available context, but offer fewer data points to aggregate.
> * **Dense Masks** (high density) provide *more aggregation points* (more "votes"), but the individual signals are *noisier* as less context is available.
>
> **Why adaptive weighting**
>
> This component simply combines the evidence from the progressive masking steps. Since sparse masks provide cleaner signals, **Adaptive Weighting** strategically gives more weight to the "votes" from these earlier, sparser steps using inverse-step weighting. This prioritizes the most reliable signals while still incorporating evidence from all scales.
>
> **Effect of hyperparameters**
>
> Our hyperparameters ($T=16$ steps, $\alpha_{min}=5\%$, $\alpha_{max}=50\%$, $N=128$ subsets, $m=10$ token subset size) were chosen to balance this trade-off effectively. The high-level idea is that $m=10$ is small enough to isolate the *consistent membership signal* from *sparse domain signals*, while $T=16$ and $N=128$ provide enough "votes" for a robust final score. We will also **add a new, comprehensive hyperparameter sensitivity analysis to the appendix**. This new section will include plots showing SAMA's AUC as $T$, $N$, and $m$ are varied, confirming these trade-offs and justifying our chosen values as a strong configuration.
>
> In short, **Subset Aggregation** first *isolates* the weak membership signal by filtering out domain noise, and **Progressive Masking** then *amplifies* this signal by combining reliable evidence from multiple, independent probes. We will revise Section 3 to make this intuition clearer.

---

> ### Author Response · Authors · 2025-11-26
>
> ### W2: Additional Defensive Strategies
>
> > "It would be good to propose or evaluate more defensive strategies to improve practical applicability."
>
> Thank you for the suggestion. Our paper provides a detailed analysis of LoRA and DP-LORA as defenses in Appendix D.5. We will **add** a state-of-the-art **training-time defense** (SOFT [1]) and an **inference-time defense** (Temperature scaling) as D.5.3 and D.5.4 in our revision.
>
> [1] Zhang et al. "SOFT: Selective Data Obfuscation for Protecting LLM Fine-tuning against Membership Inference Attacks." USENIX Security 2025.

---

### Meta-Review · Area_Chair_asUf · 2025-12-28

**Summary:**

This paper studies MIA for diffusion LMs (DLMs), a setting that is unexplored in prior work. DLMs introduce unique challenges and opportunities for MIA, particularly due to masked token prediction: the ability to probe many independent masks substantially increases the likelihood of detecting memorization. Built on this, the paper proposes SAMA, an attack that aggregates membership signals across progressively sampled masked subsets using robust sign-based statistics. Experiments on non datasets demonstrate that SAMA achieves significant improvements over baselines in both AUC and low-false-positive-rate.
Reviewers agreed that the problem introduced in the paper is new and original (9XsJ, 7q5n, VbcU, aFkU), the problem formulation, experiments, and discussion are comprehensive and insightful (9XsJ, 7q5n, aFkU), the method is sound (9XsJ, VbcU), and the paper is well written (7q5n, VbcU. aFkU).

The most significant concern raised by reviewers is that the proposed attack appears more computationally expensive than baselines (9XsJ, 7q5n, VbcU), but the authors clarified that the cost is equivalent.

Other concerns that were weight adequately addressed, or do not appear significant from the AC’s perspective.
- The attack assumes grey-box access (9XsJ, VbcU): Authors responded why this assumption is still broadly applicable; the AC agrees and thinks it is the common assumption in the MIA literature.
- DLLMs are still not widely used (9XsJ): The authors argued that the work is forward-looking and the problem of MIA for DLLMs is still worth exploring early, as DLMs are a rapidly advancing alternative to autoregressive models.
- The proposed method is specialized for DLLMs (7q5n). This is inherent to the paper’s goal.
- The intuition behind the method remains abstract (7q5n) / The method is somewhat complex (aFkU): Authors’ responses addressed them, and other reviewers explicitly noted that the method is technically sound.
- Lack of theoretical guarantee (VbcU) / Missing defenses (aFkU): These are typically not required for papers proposing new attacks.

**Reviewer Concerns:**

Noted above

**Reviewer Scores:**

Noted above

---

### Decision · Program_Chairs · 2026-01-26

Accept (Poster)